# The Paf1 complex factors Leo1 and Paf1 promote local histone turnover to modulate chromatin states in fission yeast

Laia Sadeghi[1], Punit Prasad[1], Karl Ekwall[1,†,*], Amikam Cohen[2,†,**] & J Peter Svensson[1,†,***]

## Abstract

The maintenance of open and repressed chromatin states is crucial for the regulation of gene expression. To study the genes involved in maintaining chromatin states, we generated a random mutant library in *Schizosaccharomyces pombe* and monitored the silencing of reporter genes inserted into the euchromatic region adjacent to the heterochromatic mating type locus. We show that Leo1–Paf1 [a subcomplex of the RNA polymerase II-associated factor 1 complex (Paf1C)] is required to prevent the spreading of heterochromatin into euchromatin by mapping the heterochromatin mark H3K9me2 using high-resolution genomewide ChIP (ChIP–exo). Loss of Leo1–Paf1 increases heterochromatin stability at several facultative heterochromatin loci in an RNAi-independent manner. Instead, deletion of Leo1 decreases nucleosome turnover, leading to heterochromatin stabilization. Our data reveal that Leo1–Paf1 promotes chromatin state fluctuations by enhancing histone turnover.

**Keywords** Epe1; heterochromatin; histone turnover; Paf1C; transcription
**Subject Category** Chromatin, Epigenetics, Genomics & Functional Genomics

## Introduction

Boundaries between different chromatin states must be maintained for stable gene expression patterns [1,2]. Although many different chromatin states have been described, the two most fundamental categories are active euchromatin and silent heterochromatin [3]. Constitutive heterochromatin is associated with H3K9me2/3, HP1, and low histone turnover [4,5]. Although generally inactive, heterochromatin may be transcribed during defined periods of the cell cycle, but the resulting transcripts are degraded [6–8]. The fission yeast *Schizosaccharomyces pombe* uses several alternative heterochromatin formation pathways in different regions that may substitute for one another. The RNAi pathway, which involves the proteins Dcr1 and Ago1, is the predominant mechanism used to nucleate heterochromatin [9,10]. RNAi-independent heterochromatin formation depends on transcription and RNA surveillance by factors such as Mlo3-TRAMP [11]. The constitutive heterochromatin regions in *S. pombe* are large and consist of well-defined regions where nucleation occurs. The heterochromatin subsequently spreads from these regions [12], first by propagation of the H3K9me2/3 mark and then via the binding of HP1$^{Swi6}$ and deacetylation of H4K16 by Sir2 [13,14]. The binding of HP1$^{Swi6}$ is dynamic. As RNA polymerase II (RNAPII) transcribes through H3K9me2, HP1$^{Swi6}$ sequesters the nascent RNA and delivers the RNA to the TRAMP component Cid14 for degradation [8,15]. HP1$^{Swi6}$ binds to H3K9me2/3 and recruits the H3K9 methyltransferase Clr4 to act on adjacent nucleosomes. This results in a chain reaction with self-propagation of heterochromatin via histone modification.

The spreading of heterochromatin stops at any of the defined boundary elements. In *S. pombe*, the tRNA barriers that surround the pericentric repeats, the inverted repeats (*IR*) that surround the mating type locus and most centromeres, and the rDNA at the telomeres on chromosome 3 are three different examples of boundaries between euchromatin and heterochromatin. However, the boundaries of facultative heterochromatin islands and telomeres are less well understood. TFIIIC is associated with most boundaries, including both RNAPIII-transcribed tRNAs and RNAPII-transcribed *IR* elements [16]. Several factors have been identified that maintain the chromatin boundaries at *IR* elements (notably Epe1, Mst1, and Bdf2) [17,18]. Epe1 physically interacts with Bdf2, which in turn binds to the acetylated H4 tails and protects them from deacetylation by Sir2 [18]. H4K16ac is added by Mst1 [18], and H4K16ac and H3K9ac are removed from the heterochromatin by Sir2 [14,19,20]. To date, Epe1 is the only protein that has been described to possess anti-silencing activity in *S. pombe* chromatin. Epe1 is recruited to heterochromatin [21] but is degraded from the interior through Ddb1 [22]. However, Epe1 also plays a role at the heterochromatin boundaries that is independent of H4K16ac and RNAi [18,23].

1 Department of Biosciences and Nutrition, Karolinska Institutet, Huddinge, Sweden
2 Department of Microbiology and Molecular Genetics, IMRIC, The Hebrew University-Hadassah Medical School, Jerusalem, Israel
  *Corresponding author. Tel: +46 852481039; E-mail: karl.ekwall@ki.se
  **Corresponding author. Tel: +972 26758630; E-mail: amikamc@ekmd.huji.ac.il
  ***Corresponding author. Tel: +46 852481154; E-mail: peter.svensson@ki.se
  †These author contributed equally to this work

In addition to fission yeast, RNAi-mediated heterochromatin formation has also been described in plants and *Drosophila*. However, other pathways of heterochromatin formation appear to play a role in metazoans [24]. Alternative mechanisms of heterochromatin nucleation also exist in *S. pombe* [15,25–28]. For example, facultative H3K9me2/3 heterochromatin formation at meiotic genes is RNAi-independent and requires transcription and a complex consisting of Red1, Rrp6, and Mmi1 [29–32]. These H3K9me2/3 islands are dynamically regulated in response to environmental signals [30]. Red1 is a cofactor for both RNAi- and exosome-mediated RNA degradation and degrades meiosis-specific mRNAs [33]. This type of silent chromatin is characterized by the presence of both H3K9me2/3 and HP1$^{Swi6}$. Additional distinct types of silent chromatin form over developmental genes and retrotransposons ("HOODs") [7] and the central domain of the centromere where the RNA is rapidly degraded [6,34]. In contrast to human cells in which LTR retrotransposons (HERVs) are embedded in H3K9me3-associated heterochromatin, the *S. pombe* LTR retrotransposon Tf2 is not associated with H3K9me2/3 but is readily transcribed by RNAPII. However, the nascent Tf2 transcripts are targeted to the exosome independent of Red1 [35]. The siRNA pathway may take over in the event of failure of these posttranscriptional mechanisms to degrade the Tf2 transcripts, leading to the formation of H3K9me2 [7]. Heterochromatin is strictly confined to the HOOD regions even in the absence of known boundary elements.

There are clearly gaps in our knowledge in terms of how a boundary controls the spread of chromatin states, especially in the absence of tRNA barriers. In this study, we set out to uncover mechanisms that determine heterochromatin confinement. We identified Leo1 and Paf1 as factors that reverse chromatin silencing at *IR* border regions and heterochromatin islands. Both factors are part of the Paf1C. Additionally, we found that Leo1 acted by destabilizing nucleosomes in a general manner and thereby modulating chromatin states.

# Results

## Paf1C components inhibit heterochromatin propagation across the IR-L boundary element

To identify factors required for heterochromatin boundary maintenance in *S. pombe*, we applied a random mutagenesis assay and screened the resulting mutants for the silenced reporter genes *ade6+* and *ura4+*, which were integrated at regions of normally open chromatin. The *ade6+* reporter gene was positioned at the inverted repeat boundary to the left (*IR-L*) of the silent *matK* region, whereas the *ura4+* gene was integrated 1.2 kb further into the euchromatic *matL* region (Fig 1A). The cells show no phenotype when grown on complete media (YEA or YES) under non-selective conditions. However, the cells will be resistant to 5-fluorotic acid (5FOA) when *ura4+* is silenced [36], and the colonies will be red when grown on media containing a low concentration of adenine (YE) when *ade6+* is silenced. Previously, Epe1 was demonstrated to prevent heterochromatin spread across this boundary [17]. We used the *Hermes* transposon mutagenesis system to generate mutants [37] (Fig EV1A) and isolated a clone that was both 5FOA-resistant and red on low-adenine (YE) plates; thus, this mutant had an

extended silenced region surrounding the *matK* region (Fig 1B). Upon sequencing of the DNA flanking the *Hermes* integration site, the clone was found to contain the transposable element inserted within the *SPBC13E7.08c* (*leo1+*) open reading frame 507 nucleotides from the start codon. We confirmed a single integration site of the Hermes retrotransposon by Southern blotting (Fig EV1B). As expected, the heterochromatin mark H3K9me2 was found in the *IR-L* and *matL* regions in the *leo1Δ* and *leo1::Hermes* strains as determined by chromatin immunoprecipitation (Fig 1C), demonstrating that the effect was due to *de facto* heterochromatin formation and not merely posttranscriptional degradation of *ura4+* and *ade6+*. To rule out secondary effects of the transposon integration, we constructed an HA-tagged version of the truncated Leo1$^{1–169}$ protein by PCR-directed mutagenesis of the *leo1+* gene, thereby mimicking the transposon-induced truncation. Also this strain had H3K9me2 at the *IR-L* and *matL* regions. Two regions are conserved between the *S. pombe* and human Leo1 proteins that correspond to amino acids (aa) 16–46 and 87–253 in *S. pombe* Leo1 (Fig 1D). The Leo1$^{1–169}$ protein contains a conserved N-proximal domain that may be responsible for H3 binding and almost the entire conserved Paf1 interaction domain (aa87–176) [38]. *Schizosaccharomyces pombe* Leo1 is physically and functionally associated with Paf1 [39]. The C-proximal conserved region at aa176–253 has an unknown function. Interestingly, the disrupted *leo1+* alleles (*leo1::Hermes* and *leo1$^{1–169}$::HA*) exhibited a severe growth defect not present in the full gene deletion (*leo1Δ*), suggesting a dominant effect of the truncated protein. The minor phenotypic effects of *leo1Δ* on growth and morphology were noted previously [39]. The strain with the *leo1::Hermes* allele and the strain with the truncated Leo1$^{1–169}$ grew more slowly than the WT and *leo1Δ* (Fig 1E) and also displayed a distinct cellular morphology (Fig 1F).

## Leo1 and Paf1 mediate chromatin state fluctuation but not the entire Paf1C

Leo1 is a conserved protein that participates in transcription elongation by RNAPII. The protein is a component of the similarly conserved RNA polymerase II-associated factor 1 complex (Paf1C) [40,41]. In *S. pombe*, Paf1C is minimally composed of four subunits: Paf1, Leo1, Cdc73, and Tpr1 [39] (Fig 2A). Given the role of Leo1 in Paf1C, we compared the effects on *IR-L* silencing following the deletion of genes encoding the other components of the protein complex. In addition to *leo1Δ*, *paf1Δ* enhanced silencing within *IR-L* and extended the silent domain into the *matL* region. However, deletion of *tpr1+* or *cdc73+* had little or no effect on chromatin silencing (Fig 2B). These observations demonstrate that the Leo1–Paf1 heterodimer rather than the complete Paf1C prevents the propagation of the repressed state across the *IR-L* boundary element.

## Stability of the chromatin state

We noted irregular patterns that manifested as large and small colonies on 5FOA and red-white sectoring on the low-adenine plates for the *leo1Δ* and *paf1Δ* clones (Figs 1B and 2B), suggesting an unstable chromatin silencing phenotype. The propagation of the repressed state across *IR-L* was observed in only a subset of cells within the colony. This phenotype has previously been reported in *epe1* mutants [17] and is caused by position effect variegation (PEV) of

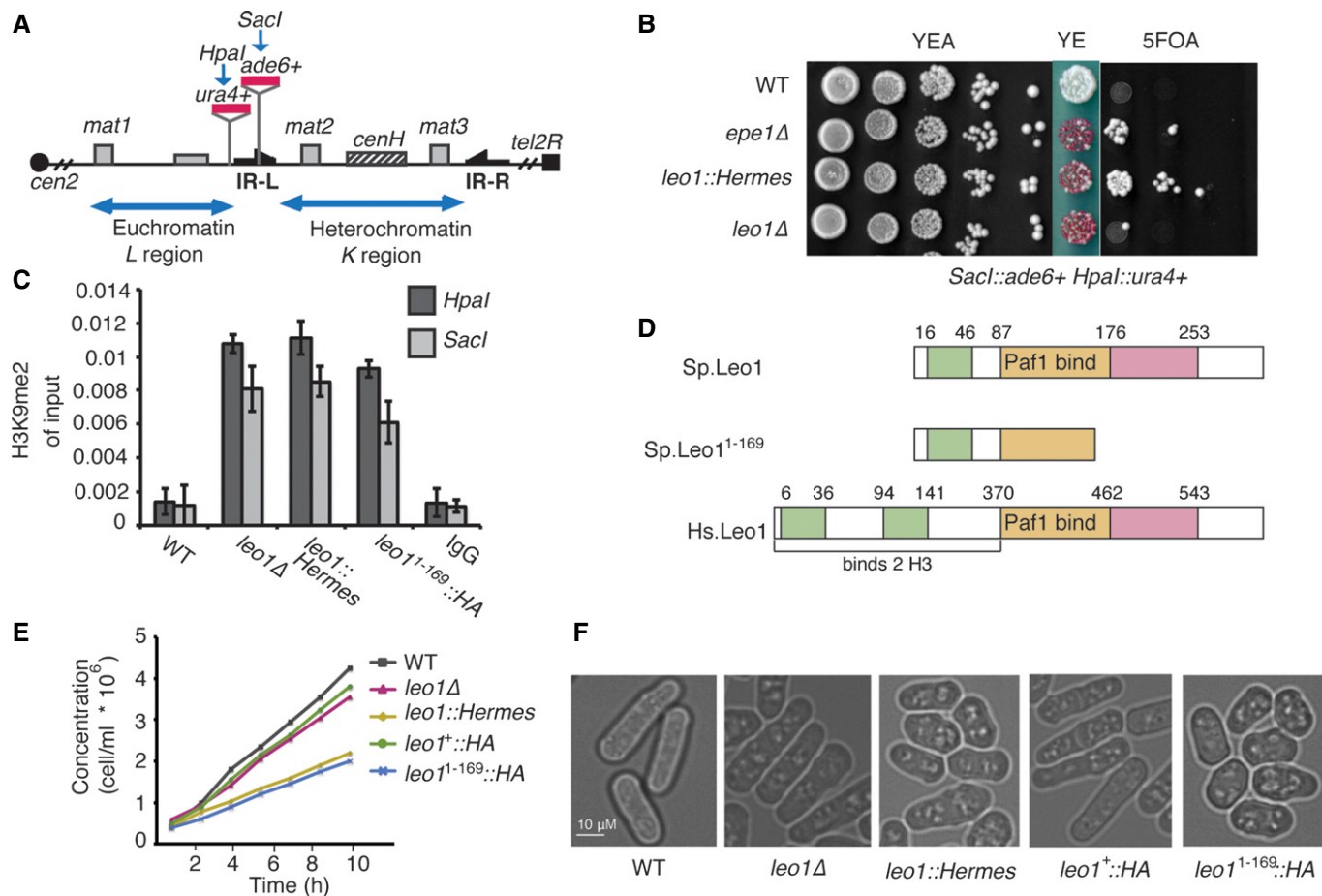

**Figure 1. Identification of Leo1 in a screen for factors counteracting heterochromatin spreading across the *IR-L* boundary.**

A   Scheme of the mating type region in *Schizosaccharomyces pombe*. Integration positions of reporter genes (*ura4⁺* or *ade6⁺*) are indicated.

B   Spotting assay using strains with *ura4⁺* and *ade6⁺* integrated as in (A) on plates with added adenine (YEA), with low adenine (YE), or with 5-fluorotic acid (5FOA).

C   ChIP–qPCR analysis of H3K9me2 levels at the *ura4⁺* or *ade6⁺* insertion sites depicted in (A). ChIP was performed in three independent experiments (*n* = 3); error bars indicate the standard deviation (SD).

D   Schematic representation of the protein domains of Leo1; Sp: *S. pombe*; Hs: human. The different colors represent conserved domains: orange: Paf1 interaction; green: conserved, potentially H3 interacting; pink: conserved region of unknown function, potentially RNA binding.

E   Growth curves of WT cells and cells with different *leo1⁺* alleles (as indicated).

F   Bright-field microscopy images of the indicated strains.

the inserted *ade6⁺* or *ura4⁺* reporters. To test whether variegation could also be observed in the Leo1–Paf1 mutants, the cells were grown on non-selective medium following selection (–Ura) and counterselection (5FOA) and then restreaked onto selective plates (Fig 2C). The results showed that once established, the repressed state of *ade6⁺* and *ura4⁺* in the *matL* region was stably maintained under non-selective conditions for more than 30 generations in the *leo1* and *paf1* mutant backgrounds. These results demonstrate that Leo1 and Paf1 (similar to Epe1) function to destabilize the repressed chromatin state.

## The effects of mutations that affect histone modifications and RNA processing

In addition to the intrinsic function of Paf1 and Leo1 in promoting RNAPII-mediated chromatin transcription [40], these components of the Paf1 complex are involved in multiple activities that facilitate

transcription elongation. In budding yeast, Paf1C interacts with Set1 to methylate Lys4 on histone H3 and with the ubiquitin ligase Bre1 to monoubiquitinate Lys120 on histone H2B [42,43]. In *S. pombe*, Set1-mediated H3 methylation is conserved, and the corresponding H2B residue Lys119 is monoubiquitinated by the Bre1 homologs Brl1 and Brl2 [44,45]. We confirmed a previous observation that H2Bub1 was partially dependent on Leo1 because H2Bub1 was reduced in *leo1Δ* (61% of WT levels), albeit not to the same extent as the H2Bub1-deficient *htb1*-K119R (Fig 2D).

To determine whether *paf1Δ* and *leo1Δ* indirectly caused heterochromatin propagation across *IR-L* by impairing H2Bub1 or H3K4me, we examined cells with deficient H2B ubiquitination (*rfp1Δ*, *shf1Δ*, and *htb1*-K119R) or H3K4 methylation (*set1Δ*) with regard to chromatin repression (Fig 2E and F). In contrast to *paf1Δ* or *leo1::Hermes*, the mutations negatively affecting H2Bub1 or H3K4me did not induce gene silencing across the heterochromatin boundary at *IR-L*. This result is in agreement with a functional

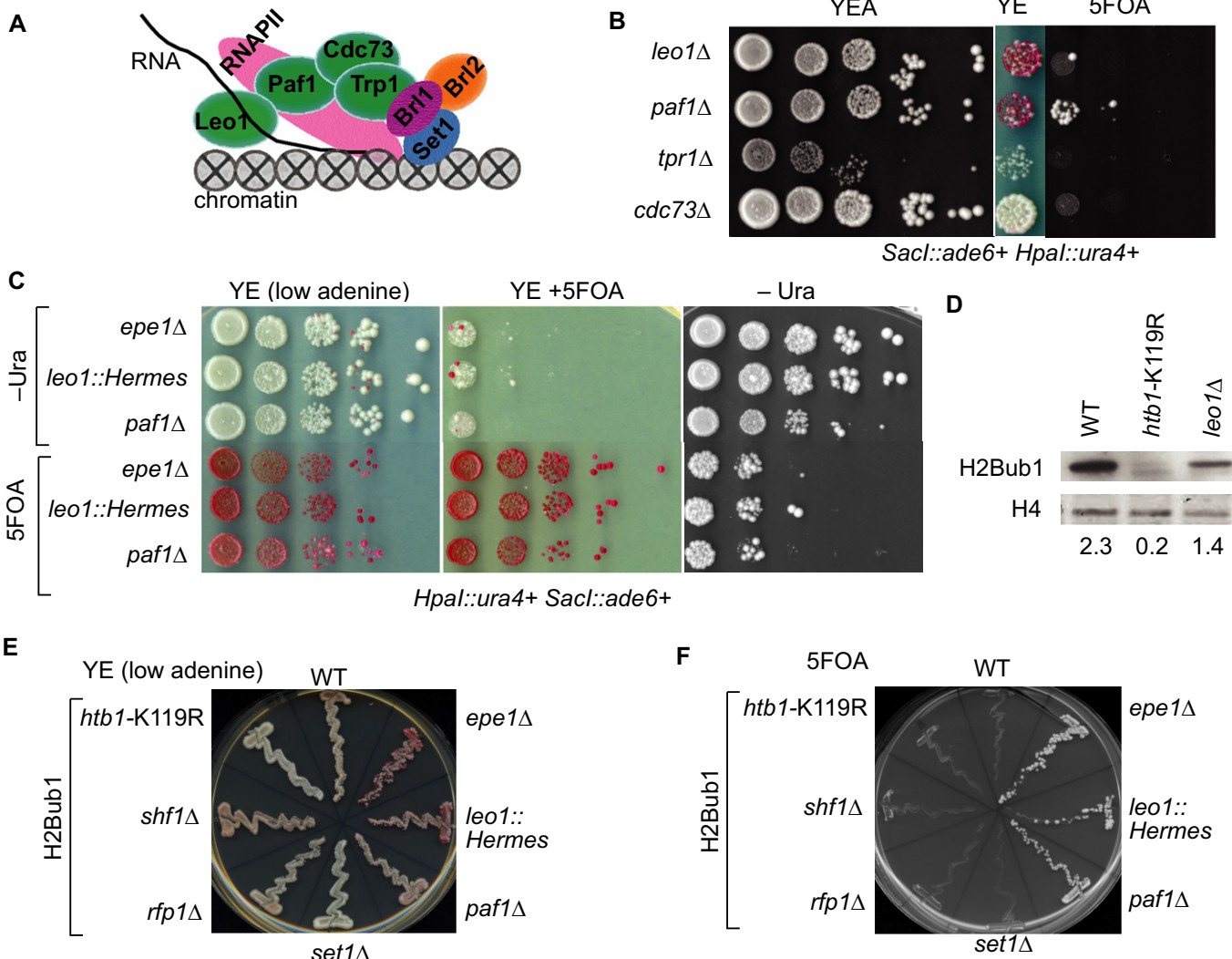

**Figure 2.  Spreading of heterochromatin across *IR-L* is specifically affected by Paf1 and Leo1.**

A    Model of Paf1C with its subunits and interactors.

B    Spotting assay using four strains with *ura4+* and *ade6+* integrated as in Fig 1A, carrying deletions of genes encoding different subunits of the Paf1 complex (as indicated).

C    Position variegation effect by Leo1 and Paf1. Spotting assay of strains regrown from –Ura or 5FOA plates with the relevant genotypes indicated.

D    Immunoblot of H2Bub1 and H4 levels in WT, *htb1*-K119R, and *leo1Δ* cells. Numbers indicate the H2Bub1/H4 ratio for each lane.

E, F  Growth assay performed on a low-adenine plate (YE) (E) and a 5FOA-containing plate (F) using mutants with *ura4+* or *ade6+* integrated as in Fig 1A. The mutants in this panel lack different factors in the H3K4me (*set1Δ*) or H2Bub1 (*rfp1Δ*, *shf1Δ*, or *htb1*-K119R) pathways.

separation between the Leo1–Paf1 heterodimer and the rest of the Paf1 complex containing Rfp1 [39]. Together, these data indicate that although Paf1C is involved in recruiting factors to generate H2Bub1 and H3K4me, the role of the Leo1–Paf1 subcomplex in heterochromatin spreading across the *IR-L* boundary is distinct from the functions of its known interactors. Thus, heterochromatin propagation is not related to the transcriptional marks H2Bub1 and H3K4me.

**Leo1–Paf1 participates in Epe1-mediated anti-silencing activity that counteracts heterochromatin assembly**

Because *leo1::Hermes* and *paf1Δ* promote silencing propagation across *IR-L* in a similar manner as *epe1Δ* (Fig 1B) and Epe1 acts as

an anti-silencer [17,21,23], we hypothesized that Epe1 interacted genetically with the two Paf1C components to impair gene silencing. The Cul4-Ddb1^Cdt2 ubiquitin ligase promotes the degradation of Epe1 within heterochromatin domains [22]. Consequently, in *ddb1Δ* cells, Epe1 is not targeted for degradation and thus is stabilized. This stabilized Epe1 impairs heterochromatin silencing [22]. To test whether Epe1-dependent expression in the *ddb1Δ* mutants was dependent on Leo1 and Paf1, we determined the effects of Epe1 stabilization on *paf1Δ*- and *leo1Δ*-dependent *ura4+* silencing in the *matK* region. As in previous reports, we concomitantly deleted *ddb1+* and its other known target *spd1+* as not to confound data by cell cycle defects caused by stabilized Spd1 [22]. The silencing of this region was enhanced by both *paf1Δ* and *leo1Δ* as determined by

growth on 5FOA (Fig 3A) and by measuring the levels of the heterochromatin mark H3K9me2 in the *leo1Δ ddb1Δ* strain at *matK* (Fig 3B) and the pericentric *dhI* region (Fig 3C) by ChIP. Endogenous *ura4*[+] expression was not affected by the inactivation of Paf1 or Epe1 (Fig EV1C). The interaction between *epe1* and *leo1/paf1* was specific to the Leo1–Paf1 subcomplex because deletion of the other components of Paf1C (*cdc73*[+] or *tpr1*[+]) did not rescue the *ddb1Δ* phenotype (Fig EV1D). These results demonstrate a genetic interaction between *epe1* and *leo1/paf1* that cooperatively counteracts chromatin silencing.

To rule out the contributions of other Ddb1 targets, we specifically increased the protein levels of Epe1 by adding a peptide tag to Epe1 that stabilized the protein [46]. As expected, compromising the Leo1–Paf1 subcomplex by deletion of *paf1*[+] or *leo1*[+] or in cells with the *leo1::Hermes* allele reversed the silencing defect in an *epe1::GFP* strain (Fig EV1E), confirming the specific genetic interaction between *epe1* and *paf1/leo1*.

## Facultative heterochromatin islands are suppressed by Leo1 and Paf1

To determine whether Leo1 imposes a general anti-silencing effect in the genome, we performed genomewide mapping of the H3K9me2 marks in WT and *leo1Δ* cells. The results with subnucleosomal resolution were obtained through histone ChIP–exo

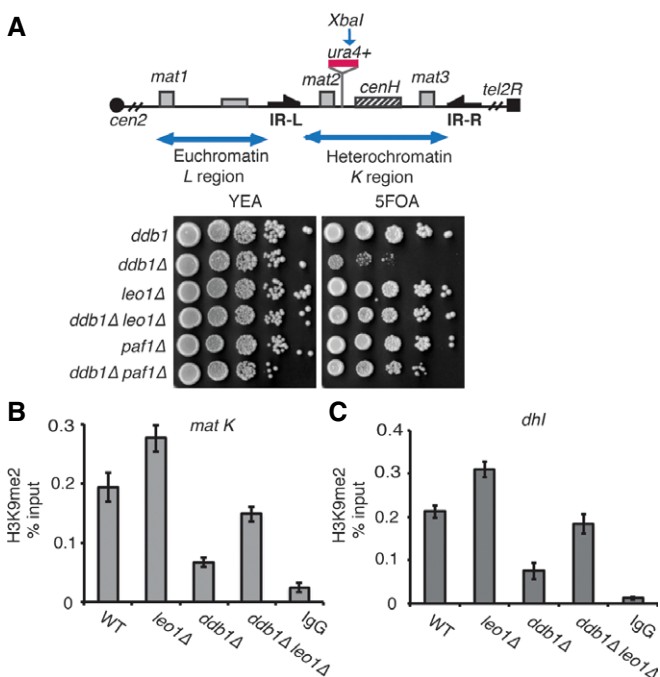

**A**

**B** *mat K*

**C** *dhI*

**Figure 3.  Leo1–Paf1 and Epe1 mediate chromatin anti-silencing.**

A  Scheme of the *mat* locus in *S. pombe*. Integration position of *ura4*[+] is indicated (upper panel). Spotting assay on YEA and 5FOA plates (lower panel).

B  ChIP–qPCR across the XbaI site in the *matK* domain.

C  ChIP–qPCR across the pericentric *dh* repeat of centromere 1.

Data information: (B, C) ChIP was performed in three independent experiments; error bars indicate SD.

[5,47]. The constitutive heterochromatin at pericentric regions was contained in the *leo1Δ* cells, but H3K9me2 spreading could be observed at the *mat* locus and at telomeres to varying degrees (Fig 4A). In addition to these constitutive heterochromatin regions, some H3K9me2-enriched regions appeared in the *leo1Δ* strain compared with the WT cells. These regions overlapped the previously identified islands of facultative heterochromatin to a large extent [30], including *mcp7*, *mei4*, and *ssm4* (Fig 4B). Most of the genes at these islands are required for meiosis [30,32].

Next, we specifically examined the retrotransposable element Tf2, which is normally posttranscriptionally silenced by exosomal degradation [7]. We observed a twofold increase and a shift in the profile of H3K9me2-marked nucleosomes at these elements in the *leo1Δ* cells (Fig EV2A). We confirmed this increase in H3K9me2 by ChIP–qPCR at Tf2s (Fig 4C). As demonstrated by the partial suppression of *leo1Δ* by *ddb1Δ* in the *matK* region above, stabilizing Epe1 by removing *ddb1*[+] (*ddb1Δ*) in the *leo1Δ* background resulted in restoration of WT levels of H3K9me2 at the Tf2s. To confirm heterochromatin formation, we also quantified the change in Swi6 association at the *Tf2* chromatin. Similar to the H3K9me2 data, the Swi6 levels at *Tf2* were also increased in the *leo1Δ* cells. This result was in contrast to the similar levels of Swi6 detected at constitutive heterochromatin (*dhI* and *matK*) and the undetectable levels at the expressed *spd1*[+] gene (Fig 4D). Silencing of the *ura4*[+] reporter gene integrated into the *Tf2-3* locus further confirmed heterochromatin formation in the *leo1Δ* background (Fig 4E). The *leo1Δ* cells also exhibited the delayed chromatin state switch characteristic of PEV at the *Tf2-3* locus (Fig EV2B).

To further characterize the contribution of Leo1–Paf1 to facultative heterochromatin assembly, we compared the effects of *leo1::Hermes* and *paf1Δ* on the expression of *ura4*[+] from an ectopic locus adjacent to *SPAC23H3.14*. Expression of this locus has been reported to be dependent on Epe1 activity [21,46]. The expression of this locus was partially dependent on Leo1 because *leo1::Hermes* inhibited the expression of *ura4*[+] from *SPAC23H3.14* (Fig 4F). Remarkably, while *leo1::Hermes* and *epe1Δ* had similar effects on gene silencing and the H3K9me2 levels at *SPAC23H3.14*, *leo1Δ* and *paf1Δ* had little to no effect on 5FOA sensitivity. To determine whether the repressive effects of *leo1::Hermes* reflected a change in chromatin structure at *SPAC23H3.14*, we conducted ChIP analysis with anti-H3K9me2 antibodies. The results show an elevated level of H3K9me2 association with *SPAC23H3.14* in the *leo1::Hermes* strain (Fig 4G).

In summary, Leo1–Paf1 (similar to Epe1) suppresses facultative heterochromatin formation at heterochromatin islands, meiotic genes, and transposable elements. Leo1 also functions to confine heterochromatin at the *mat* locus and telomeres but plays a minor role in the formation or spreading of constitutive pericentric heterochromatin.

## Paf1C regulates RNAi-independent chromatin silencing at pericentric heterochromatin domains

In contrast to heterochromatin formation at the *mat* locus and facultative heterochromatin islands, which are predominantly RNAi-independent [25,32], heterochromatin formation at the pericentric domains is severely impaired by inactivation of the RNAi machinery [9]. First, we investigated whether *leo1::Hermes* (similar to *epe1Δ*

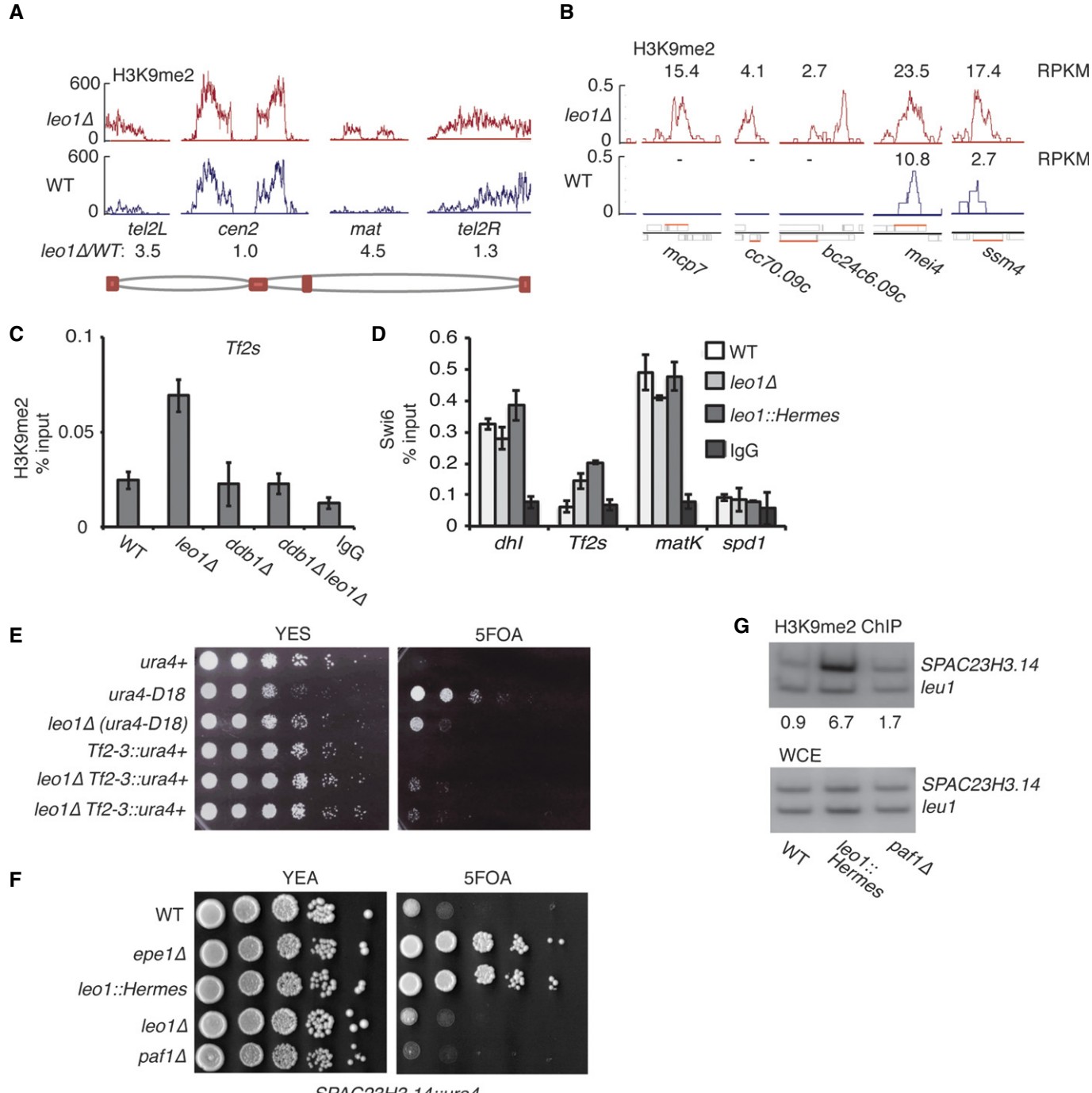

**Figure 4.  Genomewide distribution of heterochromatin in *leo1*Δ cells.**

A   A browser view of ChIP–exo signals in the WT and *leo1*Δ strains using an antibody against H3K9me2 showing the constitutive heterochromatin domains of chromosome 2 (indicated in red in the lower panel). The ratios of RPKM (reads per kilobase per million mapped reads) over the indicated region are shown. Two independent experiments were performed.

B   Browser views of H3K9me2 ChIP–exo signals in WT and *leo1*Δ cells at five facultative heterochromatin islands. The RPKM values are shown for each gene and strain.

C   ChIP–qPCR of H3K9me2 at Tf2s.

D   ChIP–qPCR of Swi6 in WT, *leo1*Δ, and *leo1::Hermes* strains at four loci.

E   Spotting assay of WT and *leo1*Δ with *ura4*+ integrated in the *Tf2-3* locus. The top row shows a WT strain with functional endogenous *ura4*+; all other strains have the non-functional *ura4-D18* allele.

F   Spotting assay using strains with *ura4*+ integrated close to the *SPAC23H3.14* locus.

G   ChIP–PCR of H3K9me2 at the *SPAC23H3.14* locus. Numbers indicate ratios for the signal *SPAC23H13.14/leu1*.

Data information: (C, D) ChIP was performed in three independent experiments; error bars indicate SD.

[4,23]) could rescue the *ago1*Δ heterochromatin defect at pericentric loci. Pericentric heterochromatin is important to ensure proper segregation of the chromosomes. As a consequence, cells that do not form pericentric heterochromatin are sensitive to the microtubule destabilizing drug thiabendazole (TBZ), as observed in *ago1*Δ cells (Fig 5A). The double mutants *ago1*Δ *leo1::Hermes* and *ago1*Δ *paf1*Δ reduced TBZ sensitivity. A related readout for pericentric heterochromatin loss is chromosome segregation defects. We can monitor the chromosome segregation process by measuring the mitotic stability of the non-essential minichromosome Ch16. The high level of Ch16 loss in *ago1*Δ cells, as determined by both the completely red colonies and red-white sectored colonies, was reversed by the *leo1::Hermes* allele (Fig 5B).

By combining the reporter gene *ura4*+ integrated at different positions in the left pericentric region of centromere 1 with *leo1* mutations, we demonstrated that the heterochromatin defects observed in the RNAi-deficient mutants were rescued by the *leo1::Hermes* allele as heterochromatin was restored (Fig 5C). Interestingly, the complete deletion of *leo1*+ only resulted in a partial rescue, again stressing the previous observation that the truncated version of Leo1 had a more severe phenotype (Fig EV2C). Surprisingly, *leo1::Hermes* and *epe1*Δ did not restore the *ago1*Δ heterochromatin defect at the *otr1R(Sph1)* locus (Fig EV2D) in contrast to the results from the left pericentric region.

In contrast to the independence of the RNAi machinery, the loss of Leo1 did not bypass the requirement for other complexes (i.e., HP1 proteins, SHREC, or FACT), as determined by crossing the mutants with *swi6*Δ, *clr3*Δ (Fig 5D), or *pob3*Δ (Fig 5E), respectively. Thus, inactivation of Leo1 and Paf1 (similar to the inactivation of Epe1) promoted RNAi-independent heterochromatin formation.

### The distribution of small RNA in WT and *leo1*Δ cells

Next, we determined the effects of the *leo1* genotype on the distribution of sRNA. To this end, small RNA (sRNA) populations were isolated and mapped to the genome. The sRNA mapping to the pericentric *dg/dh* repeats (*otr*) constituted the fourth largest sRNA population (8.4%) in the WT cells, representing transcription in both directions. This population was severely diminished in the Leo1-deficient cells (1.3%) (Fig 6A). In both the WT and *leo1*Δ cells, a small proportion of sRNAs mapped to the subtelomeric regions and then only to the *tel1L*- and *tel2R*-containing centromere-like repeats and *tlh1/2*. Few reads mapped to the facultative heterochromatin islands or the regions surrounding the *mat* locus in either the WT or *leo1*Δ strains (Fig EV3A and B). These results are consistent with the indication that heterochromatin stabilization in *leo1*-deficient cells is not mediated by RNAi, at least at the pericentric region.

The bulk of the sRNAs (54% in WT and 78% in *leo1*Δ) were single-stranded and mapped to the rDNA locus. The relative fractions of RNAPIII-transcribed tRNA and gene populations were decreased in *leo1*Δ, possibly due to the increased rRNA. We also isolated sRNAs in an RNAi-deficient strain (*ago1*Δ) (Fig EV3A–C). The *ago1*Δ results demonstrated that only the *otr* and subtelomeric sRNAs were RNAi-dependent.

Surprisingly, because the major loci of new heterochromatin in *leo1*Δ cells were outside the pericentric regions, the most substantial changes in sRNA populations mapped to the pericentric regions, and especially to the *IR* boundaries (Fig 6B). Although sRNA

transcription from the *IRC* boundaries was affected, the spreading of pericentric heterochromatin was limited in cells without fully functional Leo1 (Fig EV3D–F), suggesting that the tRNA borders were unaffected. We tested the integrity of the tRNA boundary and found that it was intact in *leo1*-deficient cells (Fig EV3G). The right pericentric border of centromere 1 lacks tRNAs and relies solely on the *IRC* element [29,48]. A slight increase in H3K9me2 at this *IRC1R* element was observed (Fig EV3H).

The fact that the pericentric heterochromatin levels were not reduced in *leo1*Δ but conversely were somewhat augmented in the *leo1::Hermes* strain (Fig EV3I) indicated that heterochromatin formation is intact, by RNAi or an alternative heterochromatin nucleation mechanism [15,25–28]. Additionally, nucleosomes in heterochromatin may have been stabilized in *leo1*Δ cells independent of RNAi.

### Relationship of heterochromatin stabilization to the loss of TRAMP and exosomes

To relate to previous studies, we investigated the effect of *leo1*Δ on pathways that have been described to stabilize heterochromatin. The loss of heterochromatin in RNAi-deficient cells has been shown to be restored by the concomitant loss of the TRAMP or exosome complexes [7,11]. The sRNA profile in *leo1*Δ was reminiscent of results from cells lacking the TRAMP subunit Cid14. In *cid14*Δ cells, the sRNAs at the *dg/dh* repeats are reduced and the *IRC* sRNAs are undetectable; this phenomena is coupled with an increase in rRNAs [49]. Conceivably, we hypothesized that Cid14 may be involved in the heterochromatin stabilization in *leo1*Δ cells. Indeed, deletion of *cid14*+ rescued the *leo1::Hermes* phenotype (Fig 6C). Silencing of *ade6*+ was also observed at the *otr1* and *mat2* loci and was dependent on *cid14*+ regardless of the *leo1* genotype (Fig EV4A).

It has also previously been demonstrated that silencing double-stranded sRNAs are generated at HOODs (many of which comprised Tf2 elements) in exosome-depleted *rrp6*Δ cells [7]. We confirmed this finding, but in contrast to *rrp6*Δ cells, in WT and *leo1*Δ cells sRNAs were primarily detected in the sense direction only (Fig EV4B), suggesting that most Tf2 sRNAs in WT and *leo1*Δ were RNAi-independent degradation products. Taken together with the previously observed increase in heterochromatin at Tf2s in the *leo1*Δ cells (Fig 4C–E), these data supported the existence of an alternative mechanism for heterochromatin stabilization.

### A role for Leo1 in histone turnover

The function of Leo1 in shifting chromatin states led us to suspect that Leo1 played a role in histone turnover. To test this possibility, we investigated the effect of the *leo1* genotype on new histone incorporation into the chromatin using the recombination-induced tag exchange (RITE) assay [5,50]. In this system, the hormone β-estradiol is used to swap the epitope tag of histone H3 (*hht2*+) from HA ("old" histone) to T7 ("new" histone) using a Cre/*Lox* recombinase (Fig 7A). H3 expression is regulated under the control of its endogenous promoter. The RITE assay was used to measure histone turnover in unsynchronized WT and *leo1*Δ cells. Samples were collected before and 2 h after the induction of the genetic switch. ChIP–qPCR was performed using HA and T7 antibodies to

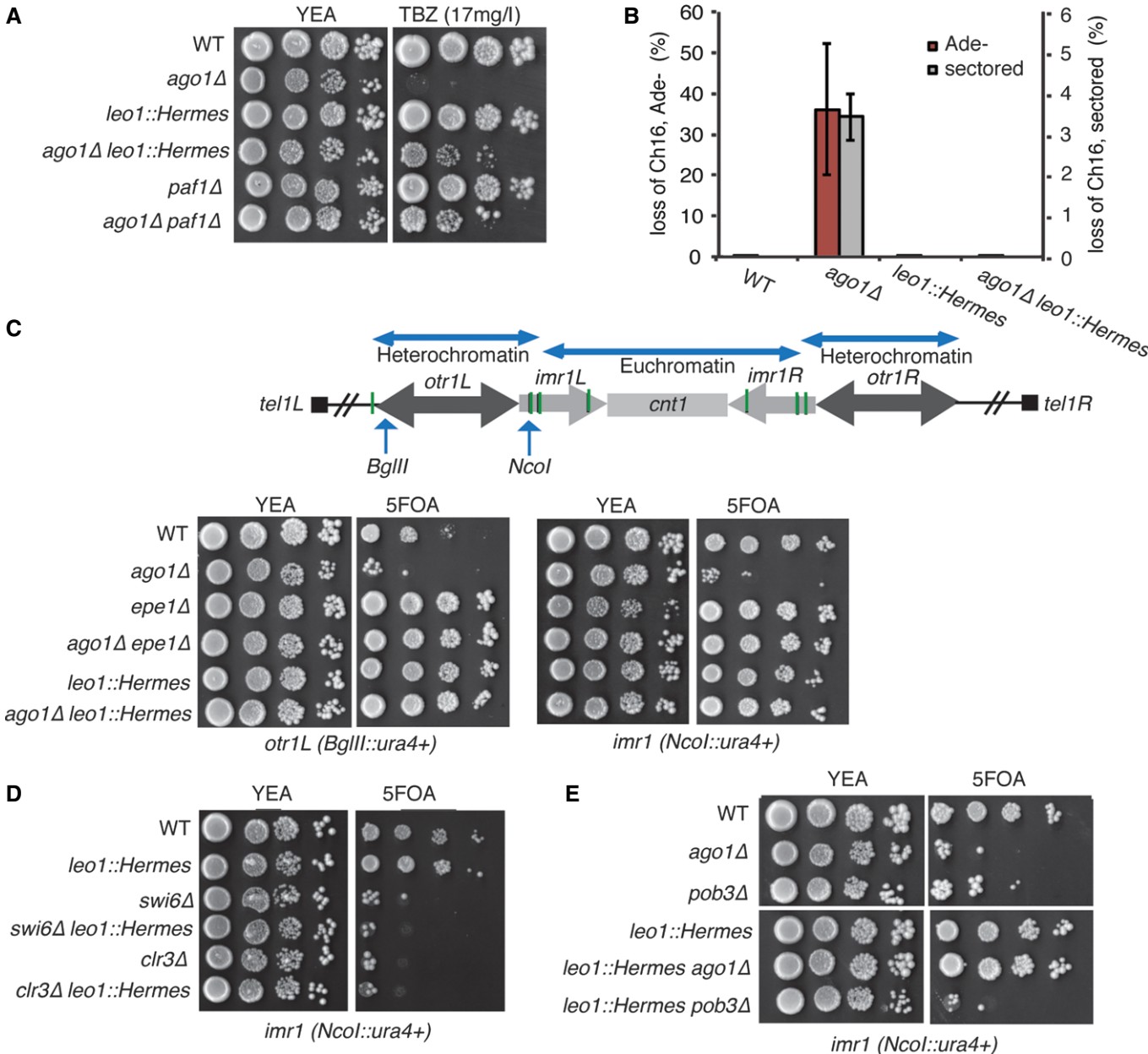

**Figure 5. Leo1 aggravates the effects of siRNA loss.**

A  Spotting assay to test TBZ sensitivity of the indicated strains. Tenfold serial dilutions of the indicated cultures were grown on rich medium (YEA) in the presence (17 μg/ml) or absence of TBZ.

B  Loss of minichromosome Ch16. Two independent experiments were performed; error bars show the range.

C  Representation of centromere 1 in *S. pombe*. Integration sites of *ura4*[+] are indicated with a blue vertical arrow (upper panel). Spotting assay of strains with the relevant genotypes indicated (lower panel).

D  Spotting of strains lacking HP1 (*swi6Δ*) or SHREC (*clr3*) with *ura4*[+] integrated at the *imr1*.

E  Spotting of strains lacking FACT (*pob3Δ*) with *ura4*[+] integrated at the *imr1*.

determine the loss of "old" histones (produced before the switch) and the incorporation of "new" histones (produced after the switch) by comparing the HA and T7 signals at 0 h and 2 h. We observed a decrease in the new incorporation of histone H3 into *leo1Δ* cells compared with the WT at the pericentric regions (*dhI*), the silent *matK* region, and the *Tf2* elements based on the lower increase in

T7 at 2 h after the genetic switch (Fig 7B). The HA signal was reduced to background levels at the heterochromatin loci *dhI*, *matK*, and *tf2* for WT and *leo1Δ*, but retention of the HA signal was observed in *leo1Δ* at the coding regions *rad50*[+] and *spd1*[+] (Fig EV4C). We also studied the effects of Ddb1 on histone turnover. As expected from the Leo1-antagonizing role we observed for

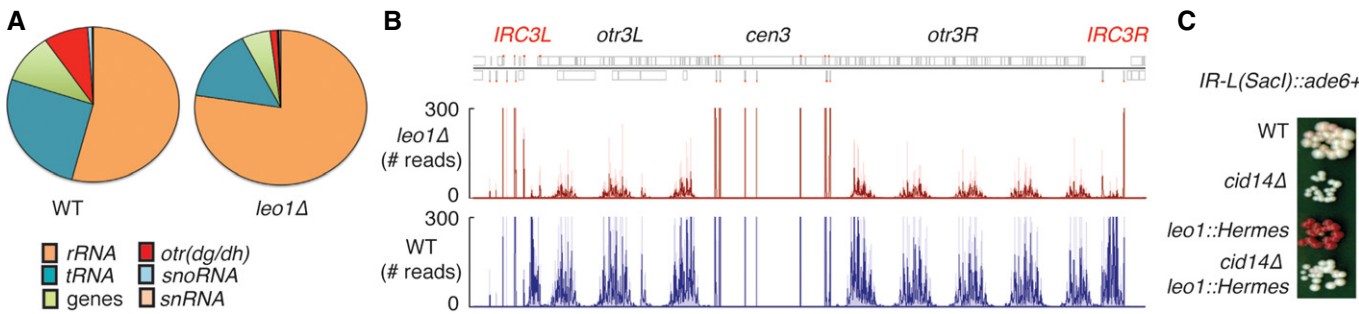

**Figure 6.  Small RNAs generated from heterochromatin are strongly reduced but still present without Leo1.**

A   Pie charts illustrating proportions between different sRNA populations in WT and *leo1Δ*.

B   sRNAs mapped across centromere 3. The number of reads from two independent experiments was normalized to the reads mapping to tRNAs. Red dots in the upper panel indicate positions of tRNAs.

C   Growth of strains with *ade6*[+] integrated at *IR-L* on a YE (low-adenine) plate.

this factor in heterochromatin formation (Figs 3A and B, and EV1), deleting *ddb1*[+] in the *leo1Δ* background indeed suppressed the histone turnover reduction in *leo1Δ*. These data clearly demonstrated a role for Leo1 in maintaining an open chromatin state by promoting histone H3 exchange.

If the mechanism whereby Leo1 elicits its chromatin effect occurs through transcription-coupled histone turnover in general, we would expect turnover to decrease at all RNAPII-transcribed loci in the *leo1Δ* cells. To this end, we determined the turnover at five additional RNAPII-transcribed loci: the boundary element *IRC1R*, *rad50*[+], *spd1*[+], *scm3*[+], and *pyk1*[+]. We also calculated the turnover at the RNAPIII-transcribed gene for the *alanine* tRNA. The results showed that the RNAPII-transcribed loci were associated with a similar reduction in histone turnover in the *leo1Δ* strain, whereas the RNAPIII-transcribed locus was not affected (Fig 7C); this result was in concordance with a general role for Paf1C in transcription elongation.

To determine whether the modification of histone turnover in and of itself could lead to the modulation of the chromatin state, we determined whether a known modifier of histone turnover (i.e., Mst2 [51]) could also influence heterochromatin propagation across the *IR-L* and *matK* loci of the *ddb1Δ* mutant. The results revealed that reducing histone turnover by deleting the histone acetyltransferase Mst2 partially restored the *matK* heterochromatin, albeit to a lower level than the deletion of Paf1 (Fig 7D). Also, heterochromatin was propagated across *IR-L*, and a PEV phenotype was observed (Fig EV4D). We also tested heterochromatin propagation when deleting another histone acetyltransferase, Hat1 [52], that has been shown to promote histone turnover in *S. cerevisiae* [53]. In contrast to *mst2Δ*, *hat1Δ* in *S. pombe* did not show a heterochromatin-spreading phenotype at *matK* or across *IR-L* (Fig EV4E and F).

### The role of transcript termination

Recently, Paf1C was shown to negatively affect siRNA-directed gene silencing through transcription termination. Mutations that inactivate Paf1C components or the Ctf1 and Res2 termination factors impair accurate transcription termination, leading to siRNA production and siRNA-mediated gene silencing in strains expressing

synthetic hairpin RNA [54]. To explore the possibility that impaired RNA processing in the *leo1::Hermes* mutants promoted heterochromatin propagation across a boundary element and suppression of the silencing defect in the *ddb1Δ* background, we determined the effects of *ctf1Δ* or *res2Δ* on the expression state of *ura4*[+] inserted at the euchromatic *matL* region or at the *matK* region of the *ddb1Δ* mutants. The results indicated that in contrast to the inactivation of Paf1, the inactivation of Ctf1 or Res2 did not promote silencing propagation across *IR-L* (Fig 7E) or the suppression of the silencing defect at the *matK* region of the *ddb1Δ* mutants (Fig 7F). These data demonstrate that faulty processing of RNA was not the cause of stabilization of the heterochromatic state in the *leo1* and *paf1* mutants observed here.

## Discussion

Chromatin state dynamics is regulated at two levels: One involves the interplay between activities that modify or de-modify nucleosomal histones, and the other involves mechanisms that enhance or suppress histone turnover. The two modes of regulation are not mutually exclusive. In fact, some proteins may function at both levels. For example, Epe1 is a putative demethylase that negatively affects H3K9 stability [55,56]. However, it may also up-regulate histone turnover at heterochromatin boundaries, leading to replacement of chromatin-associated methylated H3K9 by non-methylated H3 histones. Likewise, Mst2 may affect chromatin structure by acetylating H3K14, but it also affects heterochromatin stability by up-regulating histone turnover [51]. Paf1 and Leo1 promote chromatin transcription [41], most likely by binding H3 and destabilizing nucleosomal structure. A related activity of these Paf1C components may modulate heterochromatin spreading by up-regulating histone turnover at boundary elements. Notably, Leo1 may also antagonize heterochromatin spreading by facilitating Mst1-mediated acetylation of H4K16 [57].

### Histone turnover and stability of the heterochromatin state

Heterochromatin regions are associated with stable nucleosomes [4] that are recycled during transcription [5,58]. During replication,

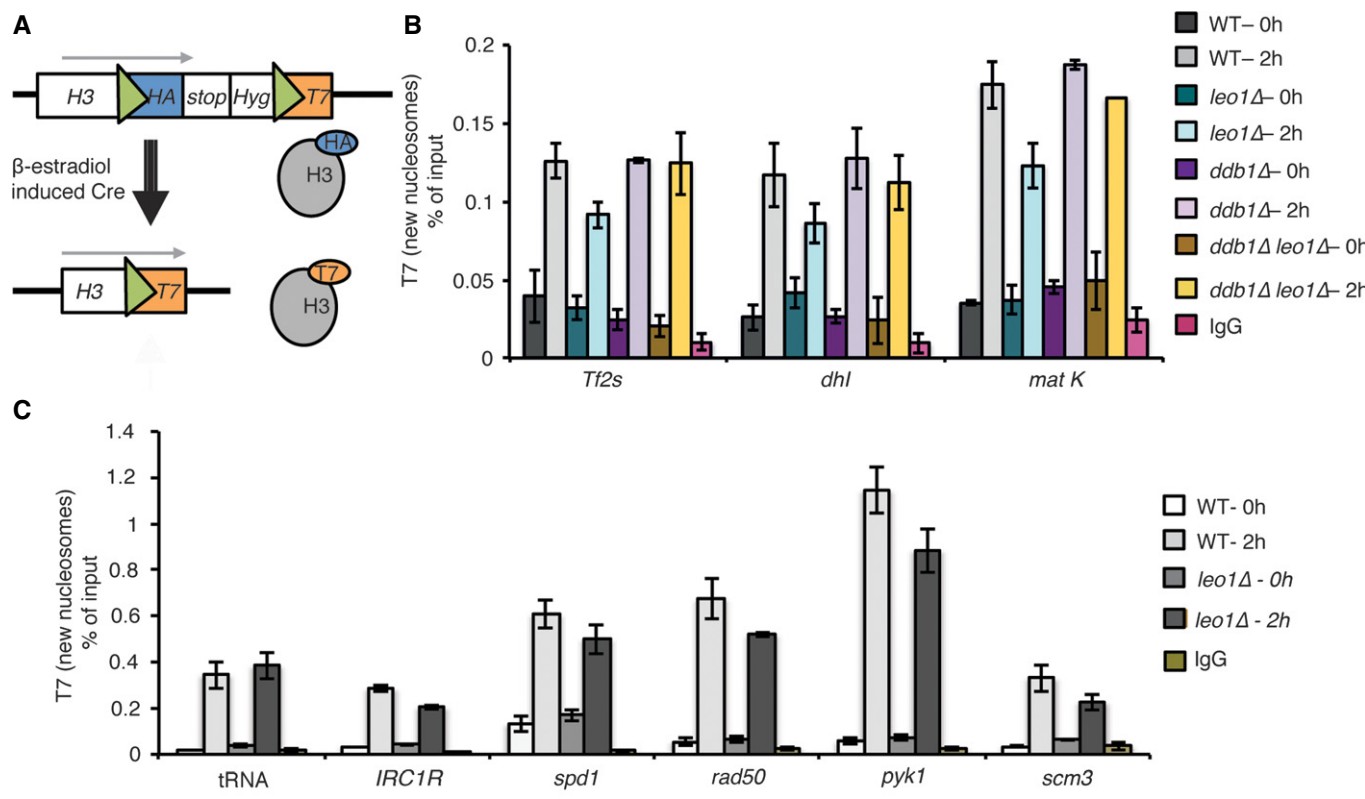

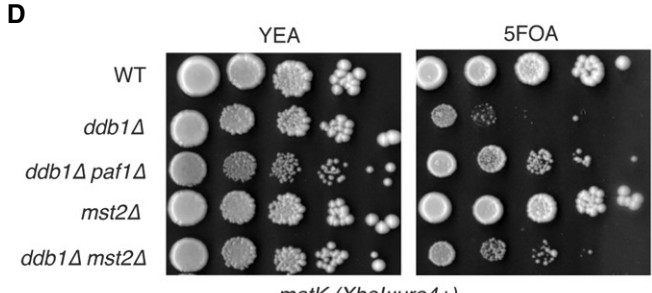

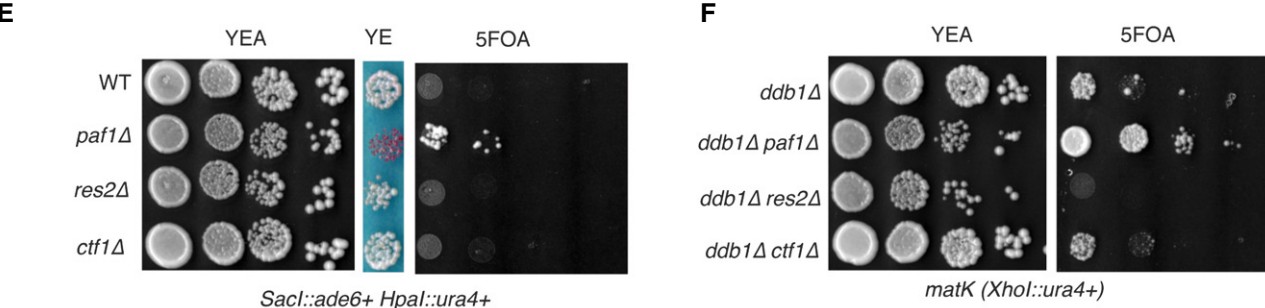

**Figure 7.  Leo1 mediates histone turnover.**

A   Diagram of the RITE system. Green triangles represent *LoxP* sites.

B   ChIP of H3-T7 at silent chromatin regions.

C   ChIP of H3-T7 at expressed euchromatic regions transcribed by RNAPII (*IRC1R*, *spd1*[+], *rad50*[+], *pyk1*[+], and *scm3*[+]) or RNAPIII (tRNA).

D   Spotting assay using strains with *ura4*[+] at the *matK* region.

E   Spotting assay using strains with *ade6*[+] integrated at the *IR-L* and *ura4*[+] integrated at *matL* domains.

F   Spotting assay using strains with *ura4*[+] at the *matK* region.

Data information: (B, C) ChIP was performed in three independent experiments; error bars indicate SD.

Swi6 associates with a Clr6-HDAC complex and HIRA, which incorporate new histones that readily acquire the posttranslational modifications of silent chromatin [59–61].

A causal relationship between up-regulation of histone turnover and modulation of heterochromatin stability has been proposed for Epe1 [4] and Mst2 [51]. The observations that inactivation of Epe1, Leo1, Paf1 and, to a lesser extent, Mst2 has an effect on propagation of heterochromatin across *IR-L* suggest that these proteins may modulate heterochromatin stability by similar mechanisms. Furthermore, *mst2Δ,* like *leo1Δ* or *paf1Δ,* suppresses silencing impairment at heterochromatin domains by *ddb1Δ.* We therefore postulate that Leo1 and Paf1, like Epe1 and Mst2, inhibit heterochromatin propagation across boundary elements by enhancing histone turnover.

The differences we observed between *mst2Δ* and *hat1Δ* have previously been observed at telomeric heterochromatin [62,63] and may be explained by the distinct mechanisms of actions by the two acetyltransferases. Hat1 acting mainly on histones in the cytoplasm before chromatin incorporation [64], whereas Mst2 belongs to the MYST family and acts on the chromatin-bound fraction [62]. This implies a distinction between (i) turnover initiated through incorporation of new histones and (ii) turnover primarily due to eviction of old histones. Consequentially, these results suggest that Leo1–Paf1 promotes turnover by eviction of histones present in the chromatin.

A direct role for Leo1–Paf1 in the destabilization of nucleosomes has previously been proposed in other organisms. Leo1–Paf1 was shown to promote chromatin transcription [40], most likely by destabilizing human nucleosomes. Leo1–Paf1 has also been shown to exhibit H3 binding affinity in human cells and *S. cerevisiae* [38]. In *S. cerevisiae*, Paf1 is required for the activation of stress-inducible genes [65], which is consistent with a role in histone exchange [5]. Other examples implicate Leo1–Paf1 in transcriptional changes during development and inflammation [66].

A role of Leo1–Paf1 in histone turnover during RNAPII-mediated transcription elongation on chromatin templates is complementary to the role of the entire Paf1C in maintaining heterochromatin by transcription termination [54]. Our data also shed additional light on the mechanism by which Leo1 was recently described to promote histone acetylation [57].

### The interaction of Leo1–Paf1 and Epe1

HP1^Swi6 recruits Epe1 to heterochromatin, where it interacts with Clr3 HDAC activity to suppress histone turnover [4]. Both *epe1* and *leo1* mutants exhibited PEV that implied defects in chromatin state fluctuations; we showed that this effect was mediated through histone turnover. Based on our observations, it is likely that Epe1 and Leo1–Paf1 act in the same pathway in this process. The question remains regarding the nature of the interaction between Leo1–Paf1 (the subunits of a transcription elongation complex) and Epe1 (a JmjC-containing enzyme). Epe1 is part of the JHDM1 family. Other members of this family act as histone demethylases [67], but the *S. pombe* homolog Epe1 lacks conserved residues within the JmjC domain [23,67]. Although no direct evidence for Epe1 demethylase activity has been reported, recent studies have proposed a putative demethylase activity [55,56].

### Relationship between Leo1–Paf1 and H2Bub1

The role of Leo1 in histone turnover together with the partial requirement of Leo1 for H2Bub1 is intriguing (Fig 2D). The reduced H2Bub1 levels in *leo1Δ* cells may be a reflection of slower RNAPII in Paf1C-compromised cells [54] because H2Bub1 is normally associated with fast elongation by RNAPII [68].

In contrast to the results obtained with the *leo1* and *paf1* mutants in this study, we previously observed the appearance of heterochromatin at the central domain of the centromere in H2Bub1-deficient cells [34]. This finding was not attributed to spreading across tRNA barriers but to *de novo* heterochromatin formation. Here, we observed that spreading across *IR* barriers was not affected by the loss of H2Bub1 (Fig 2E). H2Bub1 was directly involved in transcription elongation, and the sRNA profile of *htb1*-K119R mimicked that of *leo1Δ* (Fig EV5A and B). However, the rescue of *ago1Δ* did not depend on H2Bub1 (Fig EV5C). These observations further detail the interplay between transcription and histone turnover and the consequences for heterochromatin spread.

### Chromatin state spreading across defective boundaries

The *leo1Δ* rescue of pericentric heterochromatin in RNAi-deficient mutants (*ago1Δ*) can be explained as an indirect effect of reducing turnover and thereby stabilizing the heterochromatin formed by RNAi-independent mechanisms [69]. Because the tRNA barriers were not affected by Leo1–Paf1 (Fig EV3G), most of the pericentric heterochromatin was not challenged by the spreading of active chromatin from the surrounding euchromatin regions. In cells with defective Leo1–Paf1, only the integrity of the *IR* boundaries was compromised; thus, the surrounding euchromatin may invade into the pericentric region of *otr1R* lacking tRNAs. This is consistent with our results showing that the increased heterochromatin stability in *leo1::Hermes* cells is overruled at the right pericentric region with a defective chromatin border (Fig EV2D). Euchromatin spreading across defective boundaries has previously been described in *S. pombe* [70]. This suggests that the boundary defects we observe in Leo1–Paf1 deficient cells are bidirectional because they prevent both heterochromatin and euchromatin from spreading.

In summary, here, we present a function for Leo1–Paf1 (a subcomplex of Paf1C), which acts together with Epe1 to promote chromatin state fluctuations by enhancing histone turnover. In cells deficient in Leo1–Paf1, the increase that we observe in H3K9me2 can be explained by reduced histone turnover. Replacement of the histone H3 at the boundary element through Leo1–Paf1 or Epe1 will reverse the heterochromatin formation and maintain an open chromatin conformation. In the absence of Leo1–Paf1, Clr4 and HP1^Swi6 from the surrounding H3K9me nucleosomes propagate chromatin silencing.

## Materials and Methods

### Strains

Strains used in this study are described in Table EV1. Standard fission yeast medium was used. Growth curve was determined by

collecting exponentially growing cells at different time points. Partial deletion of *leo1* was performed by PCR-based gene targeting with G418 resistance genes as selection marker. In the spotting experiments, the dilutions were tenfold.

## Chromatin immunoprecipitation (ChIP)

Chromatin from cultures was extracted in duplicate and subjected to immunoprecipitation. Samples were immediately subjected to formaldehyde cross-linking (final concentration 1%), and chromatin was isolated according to Durand-Dubief *et al* [71]. ChIP was performed using 4 μg anti-H3K9me2 (Abcam ab1220), 1.5 μg anti-HA (Abcam ab9110), 1.5 μg anti-T7 (EMD Chemicals 69522), or 3 μg anti-Swi6 (Abcam ab188276) antibodies per 30 μl of chromatin extract. DNA was recovered with QIAquick PCR Purification columns (Qiagen). qPCR was performed on Chip samples with SYBR Master Mix (Life Technologies) using the Applied Biosystems 7500 Real-Time PCR System. A list of the primers used is provided in Table EV2.

## Western blotting

Histones were extracted using $H_2SO_4$. A total of 10 μg of histones was run on the 12% SDS–PAGE. Gel was blotted on PVDF membrane. Membrane was incubated with anti-H2Bub1 (Millipore 05-1312) (1:1,000) and anti-H4 (Abcam ab61255) (1:1,000) antibodies overnight at 4°C and detected by ECL kit.

## RITE

Cells were cultured in YES media to mid-log phase (~5 × 106 cells/ml) at 30°C. For induction of genetic switch, exponentially growing cells were treated with 1 μM β-estradiol for 2 h at 30°C.

## ChIP–exo

ChIP–exo was performed as described in Svensson *et al* [5]. *Schizosaccharomyces pombe* strains were grown in rich media and subjected to formaldehyde cross-linking and then processed through the ChIP–exo assay [72,73]. Briefly, cells were disrupted using glass beads, and chromatin was fragmented by sonication (Bioruptor Pico, 20 cycles, 30 s on, 30 s off). Fragmented chromatin was immunoprecipitated using antibodies directed against either HA or T7. Protein A-coated magnetic beads (NEB) were used to bind the antibodies. Beads were washed, and with the immunoprecipitate still on the beads, the DNA was polished, A-tailed, and ligated to an Illumina sequencing library adaptor. Digestion lambda exonuclease (final concentration 2.5 U/reaction) removed nucleotides from 5′ ends of double-stranded DNA until blocked by the formaldehyde-induced protein–DNA cross-link. The cross-links were reversed (4 h at 65°C), and DNA was eluted from the beads. The single-stranded DNA was subsequently made double-stranded by primer annealing and extension. A second sequencing adaptor was ligated. Using indexing primers, the fragments were PCR-amplified and gel-purified (Qiagen MinElute). Samples were quantified on using Qubit (HS dsDNA) and sequenced on Illumina Hiseq 2000 (50 cycles, single-end sequencing) at the BEA facility (Huddinge, Sweden) following the manufacturer's instructions. Raw data from the Hiseq (fastq files) were aligned to

ASM294v2 using Bowtie2 using default parameters. The ASM294v2.24 annotation was downloaded from pombase.org and used in Podbat. The aligned data (sam files) were imported and normalized to million reads. Data from independent biological duplicates were averaged. Identical reads were discarded to remove PCR artifacts. Signals were calculated as averages over 150 nt were taken.

## Data analysis

Genomewide data were analyzed mainly by Podbat [74] and R. RPKM (reads per kilobase per million) was calculated for all features of the ASM294v2 annotation.

## Small RNA library

Cells were grown to mid-log phase at 30°C. Total RNA was extracted with hot acid phenol and chloroform. A total of 1 μg of total RNA from each sample was used to prepare library with TruSeq Small RNA Library preparation kit (Illumina). Prepared small RNA library was sequenced using MiSeq platform at BEA facility. Small RNA libraries were sequenced on MiSeq (36 cycles, single end) at the BEA facility (Huddinge, Sweden).

## Southern blotting

Genomic DNAs from strains AP2647 (Hermes) and Hu2689 (*leo1Δ*) were purified from 50 ml of overnight cultures using standard molecular biology method. Genomic DNA (10 μg) from AP2647 and Hu2689 strains was digested with restriction enzymes EcoRI or EcoRI and KpnI for Hu2689 at 37°C overnight. The digested DNA was resolved on a 1% agarose gel for 5 h at 10 V/cm Tris–acetate–EDTA (TAE) buffer. In-gel depurination, denaturation, and neutralization were carried out before capillary transfer to Hybond-N+ membrane (0.45 μm) using 20× SSC for 20 h. The membrane was cross-linked by UV transilluminator (245 nm/1,600 J). Forward primer, SB-F (25 pmol) was end-labeled using γ-$P^{32}$ (PerkinElmer) with Optikinase (USB). PCR was carried out with the end-labeled forward primer and reverse primer (SB-R) to amplify a region within KanMx6 cassette. The probe was purified using Qiagen PCR purification kit, denatured together with 2.5 mg/ml of sheared salmon sperm DNA at 90°C for 5 min followed by snap cooling, and added to the membrane after prehybridization with buffer containing 1.5× SSC, 5× Denhardt's solution (USB), and 0.1% SDS at 64°C for 1 h. Hybridization was carried out for 18 h, and the membrane was washed with three times with 2× SSC and 0.1% SDS and exposed for phosphorimaging.

## Data availability

Data from ChIP–exo sequencing and sRNA sequencing were deposited at GEO (accession number: GSE66941).

**Expanded View** for this article is available online.

## Acknowledgements

We thank Henry L. Levin for the Hermes mutagenesis system, Ehud Sass for help and advice at early stage of this study, and Hiten D. Madhani, Robin Allshire, Danesh Moazed, Anthony T. Annunziato, Songtao Jia, and Andreas

Ladurner for strains. We thank Martin Zofall and Shiv I. Grewal for help at an early stage of this project. Funding was provided by: VR-NT, VR-M, Cancerfonden, Stockholm County Council (KE), and Åke Wiberg Foundations (JPS).

## Author contributions
LS performed experiments (ChIP, smallRNA-seq, spotting, Tf2, turnover) and wrote the manuscript; PP performed experiments (Southern blot); KE provided funding, contributed with discussing results, and wrote the manuscript; JPS performed experiments (ChIP, turnover) and wrote the manuscript; AC initiated the project, performed experiments (initial screen, most strain crosses, and spotting assays), and wrote the manuscript.

## Conflict of interest
The authors declare that they have no conflict of interest.

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
