## [Review Process File · EMBO Reports]

Manuscript EMBO-2015-41214

The Paf1 complex factors Leo1 and Paf1 promote local histone turnover to modulate chromatin states in fission yeast

Laia Sadeghi, Punit Prasad, Karl Ekwall, Amikam Cohen

Corresponding author: J. Peter Svensson, Karolinska Institutet

Review timeline:

Submission date:	18 August 2015
Editorial Decision:	31 August 2015
Revision received:	27 September 2015
Editorial Decision:	30 September 2015
Revision received:	04 October 2015
Accepted:	05 October 2015

Editor: Esther Schnapp

Transaction Report:

Transfer Note:

Please note that this manuscript was originally submitted to The EMBO Journal where it was peer-reviewed. It was then transferred to EMBO reports with the original referees' comments and the authors' response, attached. (Please see below)

Original referee's comments and author's response – The EMBO Journal

REFeree REPORTS

Referee #1:

This work begins with a genetic screen to identify mutants that fail to prevent the spreading of heterochromatin into adjacent euchromatin in *S. pombe*. The authors use a Hermes transposon-based mutagenesis approach that leads to the identification of the Leo1/Paf1 subcomplex of the Paf1 complex with a role in this process. They then use a variety of approaches including high resolution ChIP-exo to map histone H3K9me2 in leo1 mutant cells. The results show that loss of Leo1/Paf1 leads to increased heterochromatin stability at several loci. In addition, they show that loss of Leo1/Paf1 suppresses the requirement for RNAi in heterochromatin formation at pericentromeric DNA repeats, as has been shown for deletion of the Jumonji family protein Epe1. Finally, they show that deletion of leo1 decreases nucleosome turnover and suggest that Leo1/Paf1 therefore enhance chromatin state fluctuations.

The Paf1 complex and the Leo1/Paf1 module are widely conserved in eukaryotes and play broad

roles in regulation of transcription elongation, deposition of transcription-associated histone modifications, and mRNA 3' end processing. The identification of a role for the Leo1/Paf1 subcomplex in nucleosome turnover and heterochromatin containment should be of interest to specialists in the *S. pombe* chromatin field and the broader transcription/chromatin community. The paper is somewhat descriptive and does not try to provide mechanistic insight into how Leo1/Paf1 mediate histone turnover. These factors are associated with RNA polymerase II and may affect nucleosome exchange indirectly by, for example, promoting transcription-coupled histone modifications or more directly by binding to histones and disrupting nucleosomes. In my opinion, some insight into the role of Leo1/Paf1 in regulation of nucleosome stability is required to justify publication in a general readership journal such as EMBO. An analysis of the H3 binding domain of Leo1 and testing whether deletion of *leo1+* affects transcription-coupled histone modifications might help provide some insight.

Other comments.

1. There is a lot of data in the paper and the significance of some of it and how it might help support the main conclusions of the paper is unclear. For example, data in Figure 5 on the effect of *leo1* deletion on siRNA length is of uncertain significance and is largely redundant with previously published reports. Figure 6 similarly presents negative findings.
2. It is unclear that why Leo1::Hermes shows a stronger phenotype in experiments in Fig 1B and Fig 4E. This may be due to disruption of the Paf1 binding domain or disruption of the potential RNA binding domain. It may be worth constructing a truncated Leo1 that lacks the Paf1 binding domain or the RNA binding domain to probe the mechanism of the effect of Leo1::Hermes.
3. Does the Leo1/Paf1 subcomplex associate with RNA pol II in the absence of the remaining Paf1C subunits?
4. Based on the data in Figure 8, it is hard to draw the conclusion that Leo1-Paf1 function cooperatively with *pob3*. The only conclusion that can be reached based on the data is *leo1*-delete cannot rescue the effect of *pob3*-delete, but *epe1*-delete can. In any case, *leo1*-delete reduces nucleosome exchange and *pob3*-delete reduces nucleosome maintenance during transcription (opposite activities). So the reason for loss of silencing in the double mutant is unclear and requires further investigation.
5. Some controls are missing in Figure 8E. The authors should also test silencing in *pob3*-delete *epe1+*::GFP.
6. Discussion/page 11. The authors state that, "... The *S. pombe* homolog Epe1 lacks conserved residues within the JmjC-domain and the demethylase activity has not yet been reported." Is this correct? Epe1 appears to contain active site residues that are critical for catalysis and mutation of these residues have been reported to abolish the anti-silencing activity of Epe1 when the protein is not overexpressed (Trewick et al., EMBO J. 2007 November 14; 26(22): 4670-4682.).

Referee #2:

The authors investigate an important and interesting phenomenon using the tractable and well-characterized heterochromatin system of fission yeast. The manuscript describes the mechanism by which heterochromatin is contained within well-defined domains and is prevented from spreading to adjacent euchromatic regions. They identify Leo1 as a factor required to anti-silence genes placed in euchromatin near the heterochromatin-euchromatin boundary. The H3K9me3 levels at these loci spread beyond boundary elements in Leo1 mutants, which suggests that the RNA polymerase II associated Leo1-Paf1 subcomplex prevents spreading of heterochromatin into euchromatin. The authors also show that the Leo1-Paf1 subcomplex restricts heterochromatin spreading by increasing histone turnover at regions bordering heterochromatic loci. Earlier studies have shown that the anti-silencing factor Epe1 is involved in a similar process. The authors find that the Leo1-Paf1 subcomplex is involved in the Epe1-mediated histone turnover pathway and that in this pathway turnover is coupled to transcription but is independent of histone modifications. I find the manuscript to be suitable for publication in the EMBO Journal after addressing the following points.

- 1) Epe1 has been shown to mediate histone turnover by Clr3 dependent histone deacetylation. The

authors should check if the Leo1-Paf1 subcomplex mediated turnover is dependent upon histone deacetylation. If the Leo1-Paf1 subcomplex and Epe1 participate in the same pathway then why does only Epe1 mutation but not the Leo1 mutation reverse the pob3 (FACT subunit) phenotype? It appears that there are two RNAi-independent pathways for heterochromatin recruitment: one in which both Epe1 and Leo1-Paf1 participate and another in which only Epe1 participates. Please explain this in more detail.

- 2) It would be interesting to know how is RNA PolIII movement is affected in Leo1 and Epe1 mutants.
- 3) Are the levels of Swi6/Chp1/Chp2 affected in Leo1 mutants? Beside their RNAi dependent recruitment, are they also recruited in RNAi independent manner in *S. pombe*?
- 4) Figure 4A and 4B: It is hard to see what is the extent of increase or decrease in H3K9me3 at pericentric and Tel2R in leo1 mutants as compared to WT. The authors should provide numbers that represent the difference in H3K9me3 fold-enrichment between WT and the leo1 mutant.
- 5) This manuscript would be improved by making it more accessible to a wider community than *S. pombe* experts for who it appears to be written.

Referee #3:

Though a transposon mutagenesis approach, the authors of this study have identified Leo1, a component of the conserved PAF complex, as a negative regulator of heterochromatin assembly in fission yeast. They show that heterochromatin regulation depends on Leo1 and Paf1, but not other components of the PAF complex, nor specific chromatin modifiers known to be recruited by PAFc. Genome-wide analyses indicate that leo1 mutants display expansion of heterochromatin domains at the mating-type locus and at telomeres, as well as increased H3K9 methylation at known sites of facultative heterochromatin assembly including meiotic genes and transposons. Evidence is presented that the increase in heterochromatin formation is independent of RNAi, but linked to the function of the known anti-silencing factor Epe1. Histone turnover assays indicate that disruption of Leo1 is associated with reduced incorporation of new histone H3, leading to a model whereby Leo1 acts to antagonise heterochromatin assembly by promoting histone turnover.

This paper contains some very interesting observations pointing to a novel role for Leo1/PAFc in regulating RNAi-independent heterochromatin formation. However, certain aspects of the study are not that well developed in terms of mechanistic insight, and how all the observations fit together in a unified model is not entirely clear; this is not helped by the fact that in places the text lacks clarity and focus making it difficult to follow the logic of the arguments being made. In general the experiments are technically sound, but in places the conclusions drawn do not appear to be fully supported by the data (see below); in particular, I feel some form of additional evidence is required to fully support the key conclusion that Leo1 suppresses heterochromatin assembly by promoting histone turnover. That said, I think the work would be of interest to the readership of EMBO Journal, if the manuscript can be significantly revised to improve the narrative and incorporate additional experimental data as outlined below.

It should be noted that a separate study has recently reported a role for Leo1/PAFc in regulating RNAi-mediated heterochromatin assembly in fission yeast (Kowalik et al. PMID:25807481). The two studies are complementary, so I do not think this lessens the significance of the current manuscript; however, the authors should at least comment on the other study in their discussion, and ideally test whether there is any commonality in the function of Leo1 in the two systems by assessing whether mutants affecting transcription termination (which were found to affect RNAi-mediated silencing) also affect silencing at the sites analysed here.

Major issues:

1. In relation to Fig.1, it is stated in the text that a Leo11-169 allele was created to rule out secondary effects of transposon integration and that this mutant, like the hermes insertion, has a

more severe phenotype than the *leo1Δ* null. However, no data on silencing/H3K9me2 in this mutant is presented - this data should be added.

2. On p.5 it is stated that deletion of *cdc73* or *trp1* did not rescue the *ddb1Δ* phenotype, however, the data presented in Fig. EV2B are insufficient to support this conclusion as WT and *ddb1Δ* single mutant strains are not shown on the plate for comparison.

3. From the images shown in Fig. 3D I am not convinced that localisation of Epe1 is unaffected by deletion of *leo1*. This should be confirmed by chromatin-IP.

4. Fig.5 - from the small RNA sequencing data it is inferred that siRNA levels are reduced in *leo1Δ* cells. However, since it is difficult to tell whether the apparent reduction reflects a genuine decrease in siRNA levels, or simply an increase in the proportion of non-siRNA RNAs sequenced, this finding should be confirmed by northern blot.

5. Increased silencing is proposed to be RNAi-independent since no substantial increase in corresponding siRNAs is detected by small RNA sequencing - to confirm this, it should be tested whether the increased silencing is Dcr1-independent (this can be done at loci where heterochromatin maintenance does not require RNAi, such as the MAT locus).

6. The authors suggest that the decrease in siRNAs is due to a decrease in RNAPII-mediated transcription, but whether this is the case is not tested. I suggest analysing transcript accumulation in a *dcr1Δ* background, thereby avoiding the confounding effects of RNAi-mediated processing.

7. In general I found it difficult to follow the arguments made in relation to the abundance and properties of small RNAs at centromeres (p.7/fig.5C-E). Firstly, it was unclear which small RNAs were included in this analysis (those from the whole *otr*, including the IRC?). Secondly, there appears to be a small decrease in the proportion of genuine siRNAs relative to (presumably) RNA degradation products derived from *otr3* in *leo1Δ* cells, but I cannot see how the authors justify the statement that "an alternative RNAi-independent mechanism was elicited"; to determine whether these different-looking small RNAs are RNAi-independent it is necessary to test directly whether they are dependent on Dcr1. Thirdly, I could not follow whether or how the authors intended to relate these changes in small RNAs at *otr3* but not *otr2* to the differences in the elements present at the boundaries of *otr3* and *otr2*.

8. The decrease in incorporation of new H3 seen in *leo1Δ* seems relatively small, and at all but one of the loci tested could be explained as a consequence, rather than a cause, of increased heterochromatin (this is true for *Tf2*, *dhI*, *matK*, *IRC1R*, and also *rad50*, which lies just outside the right side of centromere one and therefore could be affected by heterochromatin spreading in *leo1Δ*). The authors should test some more sites that are not associated with increased heterochromatin in *leo1Δ* in order to confirm that this effect is independent of heterochromatin. Moreover, on its own I feel this data provides insufficient evidence for the key conclusion that *Leo1* destabilises heterochromatin by promoting histone turnover - this model should be tested by an alternative approach, for example testing whether other mutants affecting histone turnover (e.g. mutants of the HIR complex) also affect the stability of chromatin states, as would be predicted.

Minor issues:

1. In Fig. 1C, insufficient information is provided as to how H3K9me2 enrichment has been calculated (i.e. enrichment relative to what?)

2. On p.4, it was not clear to me what point the authors are trying to make about chromatin state stability. The authors first state that red/white sectoring in *leo1Δ* and *paf1Δ* mutants suggests unstable chromatin silencing, and then claim to test for variegation in the experiment shown in Fig. 2C, from which they infer that the silenced state can be stably maintained for more than 30 generations. However since in this assay cells taken from 5FOA (and therefore in the silenced state) were replated only on 5FOA, this assay shows only that silencing can be maintained in some proportion of cells - it does not reveal whether a proportion of cells also switch to the expressed state, i.e. whether there is variegation - to test this the cells should also be plated on -ura media.

3. Fig.2E - it would be preferable if these silencing assays were presented as serial dilutions (as in other figures) rather than streaks, as these provide a more quantitative readout.
4. Fig.3 shows that deletion of *leo1* or *paf1* rescues the silencing defect in *ddb1Δ* (i.e. Epe1 over-accumulation) cells. This is a nice experiment, but I found the written description of the experiment and the conclusion unclear (text on p.5).
5. Fig.4C - unlike at *matK*, deleting *leo1* in a *ddb1Δ* deletion background does not restore H3K9me2 at Tf2 - this difference should be discussed.
6. Fig. 4D - insufficient explanation is provided for what is shown in this figure - it is unclear what is meant by WT +/- *ura4*, and if/where the *leo1Δ* strain carries *ura4*.
7. Fig. 4E - at SPAC23H3.14::ura4, silencing is disrupted by *leo1::hermes* but not *leo1Δ* or *epe1Δ*. Is this assumed to be down to the inferred dominant negative effect of *leo1::hermes*? To confirm this, silencing in the *leo11-169* allele should be tested here.
8. Fig. EV5F, Fig. 7D - the locations of the reporter insertions should be illustrated.
9. Fig. 5F shows clearly that Cid14 is required for the silencing seen in *leo1::hermes*, but again, it is not clear what the authors are referring to when they say "the RNAi-independent transcripts found in *leo1Δ*" (p.7). The interaction with Cid14 should be further discussed.
10. Fig.7C clearly shows that *leo1::hermes* can restore silencing at the left pericentromere of chromosome in *ago1Δ* cells - is this true also for *leo1Δ*? It was not clear to me why a defect in the IRC1R boundary would explain why silencing was not restored on the right side - based on the other findings wouldn't this be expected to result in spreading of heterochromatin out into the euchromatin, rather than vice versa?
11. Fig. 8E - should the 5th row be labelled *paf1Δepe1::GFP* instead of *leo1Δepe1::GFP*?
12. I could not follow the logic of the argument relating to the FACT mutant analysis - if PAFc and FACT oppose each other in terms of incorporation of old vs new histones, would one not expect deletion of one to counteract deletion of the other, i.e. the double mutant should resemble WT?

AUTHOR'S RESPONSE

We are grateful to the three reviewers for insightful comments and suggestions on how to improve our study. As suggested by the editor of EMBO reports, we have conducted new experiments to query the link between histone turnover and heterochromatin spreading in the *leo1-paf1* mutants. All technical/presentation/clarification points are addressed below point-by-point. Two sections were highlighted by the reviewers to be clarified. These sections ("Interaction with FACT" and "Mechanisms of heterochromatin nucleation") were thoroughly rewritten in the revised manuscript. As pointed out by the reviewers, the sRNA section consisted largely of negative data and has therefore been substantially shortened and the data is mostly found in the Extended View section of the revised manuscript.

Referee #1:

This work begins with a genetic screen to identify mutants that fail to prevent the spreading of heterochromatin into adjacent euchromatin in *S. pombe*. The authors use a Hermes transposon-based mutagenesis approach that leads to the identification of the Leo1/Paf1 subcomplex of the Paf1 complex with a role in this process. They then use a variety of approaches including high resolution ChIP-exo to map histone H3K9me2 in *leo1* mutant cells. The results show that loss of Leo1/Paf1 leads to increased heterochromatin stability at several loci. In addition, they show that loss of Leo1/Paf1 suppresses the requirement for RNAi in heterochromatin formation at pericentromeric DNA repeats, as has been shown for deletion of the Jumonji family protein Epe1. Finally, they show that deletion of *leo1* decreases nucleosome turnover and suggest that Leo1/Paf1 therefore enhance chromatin state fluctuations.

The Paf1 complex and the Leo1/Paf1 module are widely conserved in eukaryotes and play broad roles in regulation of transcription elongation, deposition of transcription-associated histone modifications, and mRNA 3' end processing. The identification of a role for the Leo1/Paf1 subcomplex in nucleosome turnover and heterochromatin containment should be of interest to specialists in the *S. pombe* chromatin field and the broader transcription/chromatin community. The paper is somewhat descriptive and does not try to provide mechanistic insight into how Leo1/Paf1 mediate histone turnover. These factors are associated with RNA polymerase II and may affect nucleosome exchange indirectly by, for example, promoting transcription-coupled histone modifications or more directly by binding to histones and disrupting nucleosomes. In my opinion, some insight into the role of Leo1/Paf1 in regulation of nucleosome stability is required to justify publication in a general readership journal such as EMBO. An analysis of the H3 binding domain of Leo1 and testing whether deletion of *leo1+* affects transcription-coupled histone modifications might help provide some insight.

We agree with the reviewer that targeting the putative H3 binding-domain of Leo1 would yield very interesting results regarding the direct effects of the histone turnover mediated by the Paf1C. However, to be conclusive these lines of research would require extensive experimentation and is better suited for a new study. We have added several new experiments and textual changes to address the direct effect of Leo1-Paf1 on nucleosome turnover, most notably the comparisons with *mst2* and *hat1* (Fig 7D, EV8) and additional loci for the RITE measurements (Fig 7C). Also in the recent study from the Buhler lab (Kowalik et al 2015), they have looked at the connection between Leo1 and RNAPII (Fig 7E,F). We reference this in the revised manuscript.

Other comments.

1. There is a lot of data in the paper and the significance of some it and how it might help support the main conclusions of the paper is unclear. For example, data in Figure 5 on the effect of *leo1* deletion on siRNA length is of uncertain significance and is largely redundant with previously published reports. Figure 6 similarly present negative findings.

We have now streamlined the manuscript and removed some data less relevant for our conclusions. The negative data on the sRNA (previous Figure 5C-E and entire Figure 6) has been removed or transferred to the extended view section. Also the data on FACT and Cid14 has been shortened and better structured to focus the message.

2. It is unclear that why Leo1::Hermes shows a stronger phenotype in experiments in Fig 1B and Fig 4E. This may be due to disruption of the Paf1 binding domain or disruption of the potential RNA binding domain. It may be worth constructing a truncated Leo1 that lacks the Paf1 binding domain or the RNA binding domain to probe the mechanism of the effect of Leo1::Hermes.

Indeed, similar to what was stated above, a detailed study of different domains would give us a better mechanistic understanding of the details behind the effect we have observed. In this manuscript we only describe the protein lacking the C-terminal domain (Fig 1E,F).

A previous concern was a second integration of Hermes that might account for the affect in our strain. To rule out the possibility that the Hermes retrotransposon has integrated at other loci in addition to the *leo1* ORF, we performed a Southern blot with a Hermes probe, confirming the unique Hermes integration (EV Fig 1B).

3. Does the Leo1/Paf1 subcomplex associate with RNA pol II in the absence of the remaining Paf1C subunits?

We did not test the Paf1C association with RNAPII here. In the conserved human complex, it has previously been shown (Kim et al 2010) that Leo1-deficient Paf1C interact normally with RNAPII, whereas the Paf1-depleted shows significantly decreased binding and the complex lacking both components showed no RNAPII binding.

4. Based on the data in Figure 8, it is hard to draw the conclusion that Leo1-Paf1 function cooperatively with *pob3*. The only conclusion that can be reached based on the data is *leo1*-delete cannot rescue the effect of *pob3*-delete, but *epe1*-delete can. In any case, *leo1*-delete reduces nucleosome exchange and *pob3*-delete reduces nucleosome maintenance during transcription

(opposite activities). So the reason for loss of silencing in the double mutant is unclear and requires further investigation.

The entire FACT (pob3) section has been shortened and data has been removed, as strong conclusions could not be made.

5. Some controls are missing in Figure 8E. The authors should also test silencing in pob3-delete *epe1+::GFP*.

This figure has been removed.

6. Discussion/page 11. The authors state that, "... The *S. pombe* homolog Epe1 lacks conserved residues within the JmjC-domain and the demethylase activity has not yet been reported." Is this correct? Epe1 appears to contain active site residues that are critical for catalysis and mutation of these residues have been reported to abolish the anti-silencing activity of Epe1 when the protein is not overexpressed (Trewick et al., EMBO J. 2007 November 14; 26(22): 4670-4682.).

In the referenced paper (Trewick et al, 2007), the authors state that "rather than being a histone demethylase, Epe1 may be a protein hydroxylase that affects the stability of a heterochromatin protein[...]" However, more recently papers from the Moazed and Allshire labs (Ragunathan et al, Science, 2015; Audergon et al, Science 2015) have described Epe1 as a "putative demethylase" and presented data consistent with demethylase activity. We now refer to these papers as well but to our knowledge direct evidence of a demethylation activity has not been shown for Epe1.

Referee #2:

The authors investigate an important and interesting phenomenon using the tractable and well-characterized heterochromatin system of fission yeast. The manuscript describes the mechanism by which heterochromatin is contained within well-defined domains and is prevented from spreading to adjacent euchromatic regions. They identify Leo1 as a factor required to anti-silence genes placed in euchromatic near the heterochromatin-euchromatin boundary. The H3K9me3 levels at these loci spread beyond boundary elements in Leo1 mutants, which suggests that the RNA polymerase II associated Leo1-Paf1 subcomplex prevents spreading of heterochromatin into euchromatin. The authors also show that the Leo1-Paf1 subcomplex restricts heterochromatin spreading by increasing histone turnover at regions bordering heterochromatic loci. Earlier studies have shown that the antisilencing factor Epe1 is involved in a similar process. The authors find that the Leo1-Paf1 subcomplex is involved in the Epe1-mediated histone turnover pathway and that in this pathway turnover is coupled to transcription but is independent of histone modifications. I find the manuscript to be suitable for publication in the EMBO Journal after addressing the following points.

1) Epe1 has been shown to mediate histone turnover by Clr3 dependent histone deacetylation. The authors should check if the Leo1-Paf1 subcomplex mediated turnover is dependent upon histone deacetylation.

To address this question we have looked at the heterochromatin stabilization in a Clr3 mutant in the revised manuscript (Fig 5D). Our results show that the chromatin silencing observed in the *paf1* and *leo1* mutants depend on Clr3 as predicted by the reviewer. Further, a link between Leo1 and histone acetylation was recently presented by the Bayne lab (Verrier et al, 2015) showing that depletion of Leo1 diminishes the acetylation of H4K16. We have included this reference in the revised manuscript.

If the Leo1-Paf1 subcomplex and Epe1 participate in the same pathway then why does only Epe1 mutation but not the Leo1 mutation reverse the pob3 (FACT subunit) phenotype? It appears that there are two RNAi-independent pathways for heterochromatin recruitment: one in which both Epe1 and Leo1-Paf1 participate and another in which only Epe1 participates. Please explain this in more detail.

This section was confusing and the data not very strong so we have removed it. Now we only state that *leo1::Hermes* cannot rescue the *pob3Δ* phenotype (Fig 6E).

2) It would be interesting to know how is RNA PolII movement is affected in *Leo1* and *Epe1* mutants.

In the recent paper from the Buhler lab (Kowalik et al, 2015), the authors determined how the RNAPII occupancy was affected by mutations in *leo1* and *paf1*. Together with the finding that the elongation rate of RNAPII is increased by H2Bub1 (Fuchs et al, 2014) and our finding that the H2Bub1-levels are decreased in *leo1Δ* cells (Figure 2D), this suggests that the RNAPII is slowed down in the *leo1* mutants. We have included a section on this in the Discussion (p10, third paragraph).

3) Are the levels of Swi6/Chp1/Chp2 affected in *Leo1* mutants? Beside their RNAi dependent recruitment, are they also recruited in RNAi independent manner in *S. pombe*?

We have performed ChIP-qPCR experiments using an antibody against Swi6 in the revised manuscript (Fig 4D). The data show that as H3K9me2, also Swi6 is increased at the RNAi-independent heterochromatin in the *leo1Δ* strain.

4) Figure 4A and 4B: It is hard to see what is the extent of increase or decrease in H3K9me3 at pericentric and *Tel2R* in *leo1* mutants as compared to WT. The authors should provide numbers that represent the difference in H3K9me3 fold-enrichment between WT and the *leo1* mutant.

In the revised manuscript we have included the fold-enrichment (or absolute values for the facultative heterochromatin loci) in the WT and *leo1* mutant (Fig 4A,B).

5) This manuscript would be improved by making it more accessible to a wider community than *S. pombe* experts for who it appears to be written.

This point is well taken. We have tried to make the manuscript more accessible to a wider audience during this revision by having a clearer focus and removing much of the non-essential information. We have also had the text professionally language edited.

Referee #3:

Though a transposon mutagenesis approach, the authors of this study have identified *Leo1*, a component of the conserved PAF complex, as a negative regulator of heterochromatin assembly in fission yeast. They show that heterochromatin regulation depends on *Leo1* and *Paf1*, but not other components of the PAF complex, nor specific chromatin modifiers known to be recruited by PAFc. Genome-wide analyses indicate that *leo1* mutants display expansion of heterochromatin domains at the mating-type locus and at telomeres, as well as increased H3K9 methylation at known sites of facultative heterochromatin assembly including meiotic genes and transposons. Evidence is presented that the increase in heterochromatin formation is independent of RNAi, but linked to the function of the known anti-silencing factor *Epe1*. Histone turnover assays indicate that disruption of *Leo1* is associated with reduced incorporation of new histone H3, leading to a model whereby *Leo1* acts to antagonise heterochromatin assembly by promoting histone turnover.

This paper contains some very interesting observations pointing to a novel role for *Leo1*/PAFc in regulating RNAi-independent heterochromatin formation. However, certain aspects of the study are not that well developed in terms of mechanistic insight, and how all the observations fit together in a unified model is not entirely clear; this is not helped by the fact that in places the text lacks clarity and focus making it difficult to follow the logic of the arguments being made. In general the experiments are technically sound, but in places the conclusions drawn do not appear to be fully supported by the data (see below); in particular, I feel some form of additional evidence is required to fully support the key conclusion that *Leo1* suppresses heterochromatin assembly by promoting histone turnover. That said, I think the work would be of interest to the readership of EMBO Journal, if the manuscript can be significantly revised to improve the narrative and incorporate additional experimental data as outlined below.

To help with these issues, we have considerably edited the text for clarity and focus. Additional experiments were performed to address the reviewer's concerns.

It should be noted that a separate study has recently reported a role for *Leo1*/PAFc in regulating

RNAi-mediated heterochromatin assembly in fission yeast (Kowalik et al. PMID:25807481). The two studies as complementary, so I do not think this lessens the significance of the current manuscript; however, the authors should at least comment on the other study in their discussion, and ideally test whether there is any commonality in the function of Leo1 in the two systems by assessing whether mutants affecting transcription termination (which were found to affect RNAi-mediated silencing) also affect silencing at the sites analysed here.

Yes, we are aware of this very recent paper. In the revised manuscript, we have related our results to their findings and performed additional experiments (Fig 7E,F, text in Results p8 and Discussion). We have tested the transcription termination effect seen in the Kowalik et al study by determining the role of Ctf1 and Res2 in suppressing the anti-silencing effect in *ddb1Δ* cells at the *matK* region. The results show that the *ctf1Δ* or *res2Δ* cells do not restore heterochromatin, as does *leo1Δ* or *paf1Δ*. Therefore, we conclude that the heterochromatin stabilization effect in the *leo1/paf1* mutants that we see is not related to transcription termination, but rather represent an alternative anti-silencing effect by histone turnover.

Major issues:

1. In relation to Fig. 1, it is stated in the text that a *Leo11-169* allele was created to rule out secondary effects of transposon integration and that this mutant, like the *hermes* insertion, has a more severe phenotype than the *leo1Δ* null. However, no data on silencing/H3K9me2 in this mutant is presented - this data should be added.

We have now confirmed the unique integration by Southern blot (Figure EV1B). Due to the editor's concern for timeliness, we have not included ChIP-qPCR data for the *leo11-169* strain.

2. On p.5 it is stated that deletion of *cdc73* or *trp1* did not rescue the *ddb1Δ* phenotype, however, the data presented in Fig. EV2B are insufficient to support this conclusion as WT and *ddb1Δ* single mutant strains are not shown on the plate for comparison.

The growth of WT and *ddb1Δ* single mutant is shown in Fig 3A.

3. From the images shown in Fig. 3D I am not convinced that localisation of Epe1 is unaffected by deletion of *leo1*. This should be confirmed by chromatin-IP.

This is a good suggestion. However, due to the editor's concern regarding timeliness we have not performed this experiment, although it is technically feasible.

4. Fig.5 - from the small RNA sequencing data it is inferred that siRNA levels are reduced in *leo1Δ* cells. However, since it is difficult to tell whether the apparent reduction reflects a genuine decrease in siRNA levels, or simply an increase in the proportion of non-siRNA RNAs sequenced, this finding should be confirmed by northern blot.

This is true. We don't know if the siRNAs are reduced, and they may not be. However, this is not essential for our main conclusion that heterochromatin is destabilized by Pac1C-promoted histone turnover. What is important is that the siRNAs are not increased as would have been expected from an RNAi-dependent increase in heterochromatin. We have rewritten the text. Although this would be a nice experiment to do, in our experience Northern blots of siRNA are very cumbersome. These experiments would delay our paper significantly. We now put considerably less focus on the siRNA in the manuscript, as these were largely negative data.

5. Increased silencing is proposed to be RNAi-independent since no substantial increase in corresponding siRNAs is detected by small RNA sequencing - to confirm this, it should be tested whether the increased silencing is Dcr1-independent (this can be done at loci where heterochromatin maintenance does not require RNAi, such as the *MAT* locus).

In Figure 5A-C we show that the increased silencing of the pericentric regions is RNAi (Ago1)-independent. This point would require substantial additional experiments, that the editor suggested against.

6. The authors suggest that the decrease in siRNAs is due to a decrease in RNAPII-mediated

transcription, but whether this is the case is not tested. I suggest analyzing transcript accumulation in a *dcr1Δ* background, thereby avoiding the confounding effects of RNAi-mediated processing.

Similar experiments were performed in the Kowalik et al paper. They performed ChIP for RNAPII (Fig 3C) and also looked at the effect of Paf1C mutations on transcription (Fig S8). We refer to these results.

7. In general I found it difficult to follow the arguments made in relation to the abundance and properties of small RNAs at centromeres (p.7/fig.5C-E). Firstly, it was unclear which small RNAs were included in this analysis (those from the whole otr, including the IRC?).

Secondly, there appears to be a small decrease in the proportion of genuine siRNAs relative to (presumably) RNA degradation products derived from *otr3* in *leo1Δ* cells, but I cannot see how the authors justify the statement that "an alternative RNAi-independent mechanism was elicited"; to determine whether these different-looking small RNAs are RNAi-independent it is necessary to test directly whether they are dependent on Dcr1.

Thirdly, I could not follow whether or how the authors intended to relate these changes in small RNAs at *otr3* but not *otr2* to the differences in the elements present at the boundaries of *otr3* and *otr2*.

The results from the sRNA analysis were largely negative. We have therefore removed large parts of this section for clarity as they were not essential for this study. All three points raised by the reviewer were taken out of the manuscript.

8. The decrease in incorporation of new H3 seen in *leo1Δ* seems relatively small, and at all but one of the loci tested could be explained as a consequence, rather than a cause, of increased heterochromatin (this is true for *Tf2*, *dhI*, *matK*, *IRC1R*, and also *rad50*, which lies just outside the right side of centromere one and therefore could be affected by heterochromatin spreading in *leo1Δ*). The authors should test some more sites that are not associated with increased heterochromatin in *leo1Δ* in order to confirm that this effect is independent of heterochromatin. Moreover, on its own I feel this data provides insufficient evidence for the key conclusion that *Leo1* destabilises heterochromatin by promoting histone turnover - this model should be tested by an alternative approach, for example testing whether other mutants affecting histone turnover (e.g. mutants of the HIR complex) also affect the stability of chromatin states, as would be predicted.

This point has been heavily discussed. As strongly suggested by the reviewer, we have now incorporated data on the interaction with two histone acetyltransferases that affect histone turnover *Hat1* (in *S. cerevisiae*) and *Mst2*. *Mst2* that has been shown to directly affect histone turnover in *S. pombe* (Wang et al, 2015, Elife). When deleting *mst2*, we find a similar stabilization of heterochromatin as with defect *leo1* or *paf1* (Fig 7D). The *hat1Δ* strain show no effect (Fig EV8). We discuss these differences in detail in the Discussion section (fourth paragraph, p9). We have also performed the RITE turnover measurements at two more euchromatic loci, (*pyk1* and *scm3*). We find the same effect that the histone turnover is reduced in *leo1Δ* cells (Fig 7C).

The effect of heterochromatin formation in HIRA mutants was previously determined in the Whitehall and Grewal labs (Anderson et al, 2010; Blackwell et al 2004; Yamane et al, 2011). Contrarily to the suggestion by the reviewer, HIRA proteins actually maintain the heterochromatin and mutations create a less condensed heterochromatin. We have now included a reflection on this in the Discussion section (p9: "During replication, *Swi6* associates with a *Clr6*-HDAC complex and HIRA, which incorporate new histones that readily acquire the new before: "These results demonstrate a genetic interaction between *epe1* and *leo1/paf1* that cooperatively counteract chromatin silencing".

5. Fig4C - unlike at *matK*, deleting *leo1* in a *ddb1Δ* deletion background does not restore H3K9me2 at *Tf2* - this difference should be discussed.

Thank you for pointing this out. We had misstated the effect on H3K9me2 in the previous version of the manuscript. In the revised manuscript we state "stabilizing *Epe1* by removing *ddb1+* (*ddb1Δ*) in the *leo1Δ* background resulted in restoration of WT levels of H3K9me2" (p.6).

6. Fig 4D - insufficient explanation is provided for what is shown in this figure - it is unclear what is meant by WT +/- ura4, and if/where the *leo1Δ* strain carries ura4.

We have now clarified the *ura4*-status of the strains in Figure 4D (in figure and legend). In the legend we have added “The top row shows a WT strain with functional endogenous *ura4+*; all other strains have the non-functional *ura4-D18* allele.”

7. Fig. 4E - at SPAC23H3.14::ura4, silencing is disrupted by *leo1::hermes* but not *leo1Δ* or *epe1Δ*. Is this assumed to be down to the inferred dominant negative effect of *leo1::hermes*? To confirm this, silencing in the *leo11-169* allele should be tested here.

Yes, we argue that this is because the dominant effect of *leo1::Hermes* that we described previously (Fig 1E,F). We did not do the suggested experiment with the *leo11-169* allele as it is technically challenging to select for these double mutants. However, in the revised manuscript we have further addressed the specificity issue of the Hermes integration (having a single integration) by Southern blot (Figure EV1B). This should rule out other secondary mutations confounding our results.

8. Fig. EV5F, Fig. 7D - the locations of the reporter insertions should be illustrated.

The locations of the reporter genes have been included in the corresponding figures EV6D, EV7A, and EV4B (for completion), as suggested.

9. Fig. 5F shows clearly that *Cid14* is required for the silencing seen in *leo1::hermes*, but again, it is not clear what the authors are referring to when they say "the RNAi-independent transcripts found in *leo1Δ*" (p.7). The interaction with *Cid14* should be further discussed.

Again, thank you for pointing this out. We have changed the text to “In *cid14Δ* cells, the sRNAs at the *dg/dh* repeats are reduced and the *IRC* sRNAs are undetectable”. We have rewritten the section on *Cid14* to better reflect the results. We now put less emphasis on the *Cid14*, as the literature contains conflicting interpretations and we cannot bring clarity into the field with the results presented here.

10. Fig.7C clearly shows that *leo1::hermes* can restore silencing at the left pericentromere of chromosome in *ago1Δ* cells - is this true also for *leo1Δ*? It was not clear to me why a defect in the *IRC1R* boundary would explain why silencing was not restored on the right side - based on the other findings wouldn't this be expected to result in spreading of heterochromatin out into the euchromatin, rather than vice versa?

In contrast to *leo1::Hermes*, the *leo1Δ* only partially restore the *ago1Δ* defect at the left pericentric region (as shown in fig EV4A). This is stated in the text. In the *ago1Δ* cells, pericentric heterochromatin is not formed properly (Fig EV4B) so consequently it cannot spread. We argue that the pericentric heterochromatin rescue in *leo1/paf1* mutants is due to decreased histone turnover that stabilizes chromatin. This is only possible as long as the chromatin borders are intact. We have clarified relevant part in the Discussion.

11. Fig. 8E - should the 5th row be labelled *paf1Δepe1::GFP* instead of *leo1Δepe1::GFP*?

This panel has been removed.

12. I could not follow the logic of the argument relating to the FACT mutant analysis - if PAFc and FACT oppose each other in terms of incorporation of old vs new histones, would one not expect deletion of one to counteract deletion of the other, i.e. the double mutant should resemble WT?

We have removed large parts of the FACT results and discussion as the data was somewhat contradictory and we were not able to explain it clearly. In the current manuscript, we merely state “In contrast to the independence of the RNAi machinery, the loss of *Leo1* did not bypass the requirement for other complexes (i.e., HP1 proteins, SHREC or FACT)”.

Thank you for the submission of your revised manuscript to EMBO reports. Given that the comments by former referee 1 have basically not been addressed, I only sent the revised study back to referee 3, who also raised the most crucial concerns.

The comments of former referee 3 (now referee 1) are pasted below, and as you will see, although s/he acknowledges that the hypothesis that Leo1 destabilises heterochromatin by promoting histone turnover has been strengthened, this referee is nevertheless not convinced by the revised study and indicates that significant revisions are still required before the manuscript can be considered for publication here.

I am also a little surprised by your statement in the point-by-point response that certain experiments could not be performed due to the "editor's concern regarding timeliness". A three month period for manuscript revisions is standard for many journals, and if requested revisions cannot be performed in this time frame, manuscripts are usually rejected. We are in a slightly unusual situation because this is a manuscript transfer, and perhaps it was not made sufficiently clear which revisions should be performed. However, given now that former referee 3 is not convinced that the current data are sufficient to support the conclusions and certainly former referee 1 would not be convinced either by the revised version, we cannot offer to publish your manuscript at this point.

It appears that addressing the remaining concerns by referee 1 listed below can be done in a timely manner. Regarding point 1, the statement "data not shown" is not accepted at any EMBO press journal. Regarding point 2, it is standard to show single and double mutants side by side in rescue experiments.

The referee further notes that addressing point 3 is important, and as far as I understand you indicate that Epe1 ChIP can be done. Points 4 and 8 are easy to address. Additional comments are also easy to address. I therefore think that the remaining concerns can be addressed in a timely manner and I am willing to give you the exceptional opportunity to revise the manuscript a second time (which we usually do not allow) to address the remaining concerns.

Please note that it is EMBO reports policy that manuscripts must be accepted 6 months after a first decision was made, which was on the 30th of April 2015 in this case. This means that your manuscript must be accepted before the end of October. Otherwise, you can submit the revised manuscript as a new submission and the only difference will be that I will check the novelty at the time of submission again. I will ask referee 1 to assess the revised version in both cases.

Please let me know whether you agree to revise the manuscript a second time.

REFEREE REPORTS

Referee #1:

In the revised manuscript the authors have removed some of the original data that was difficult to interpret and contributed little to the overall story - I think this was a good idea as the focus and readability have improved as a result. The manuscript has also benefited from the addition of some new data, in particular: (1) evidence showing that the heterochromatin stabilisation in *leo1Δ* cells reported here is distinct from the enhanced RNAi-mediated heterochromatin formation recently reported in *leo1Δ* cells by Kowalik et al; and (2) further data that somewhat strengthens the key conclusion of a causal link between histone turnover and heterochromatin spreading.

However, I still have a number of concerns, including several points raised in my original review that have not been addressed and in my view are important to support the conclusions drawn.

Comments on original major points:

(1) In relation to Fig.1, it is stated in the text that a *Leo1*-169 allele was created to rule out secondary effects of transposon integration and that this mutant, like the *hermes* insertion, has a

more severe phenotype than the *leo1Δ* null. However, no data on silencing/H3K9me2 in this mutant is presented - this data should be added.

We have now confirmed the unique integration by Southern blot (Figure EV1B). Due to the editor's concern for timeliness, we have not included ChIP-qPCR data for the *leo11-169* strain.

Without either a silencing assay or ChIP-qPCR data, the statement that this mutant has a more severe phenotype remains unsupported - the data on which this statement is based must be shown.

(2) On p.5 it is stated that deletion of *cdc73* or *trp1* did not rescue the *ddb1Δ* phenotype, however, the data presented in Fig. EV2B are insufficient to support this conclusion as WT and *ddb1Δ* single mutant strains are not shown on the plate for comparison.

The growth of WT and *ddb1Δ* single mutant is shown in Fig 3A.

This is not sufficient as it is a different figure - to be able to properly assess whether *cdc3Δ* or *trp1Δ* rescue the *ddb1Δ* phenotype, the double mutants and the *ddb1Δ* single mutant must be compared side-by-side in the same assay.

(3) From the images shown in Fig. 3D I am not convinced that localisation of Epe1 is unaffected by deletion of *leo1*. This should be confirmed by chromatin-IP.

This is a good suggestion. However, due to the editor's concern regarding timeliness we have not performed this experiment, although it is technically feasible.

I think this experiment is important; based only on the images in Fig.3D I would not agree with the conclusion that deletion of *Epe1* has no effect on *Epe1* localisation.

(4) Fig.5 - from the small RNA sequencing data it is inferred that siRNA levels are reduced in *leo1Δ* cells. However, since it is difficult to tell whether the apparent reduction reflects a genuine decrease in siRNA levels, or simply an increase in the proportion of non-siRNA RNAs sequenced, this finding should be confirmed by northern blot.

This is true. We don't know if the siRNAs are reduced, and they may not be. However, this is not essential for our main conclusion that heterochromatin is destabilized by Pac1C-promoted histone turnover. What is important is that the siRNAs are not increased as would have been expected from an RNAi-dependent increase in heterochromatin. We have rewritten the text.

Although this would be a nice experiment to do, in our experience Northern blots of siRNA are very cumbersome. These experiments would delay our paper significantly. We now put considerably less focus on the siRNA in the manuscript, as these were largely negative data.

Although the authors acknowledge that they cannot say whether siRNAs are reduced or not, the text still refers to siRNAs being reduced; for example: "...low levels of siRNAs were sufficient for heterochromatin nucleation or that an alternative heterochromatin nucleation mechanism compensated for the RNAi machinery". Either all suggestion that siRNAs are reduced should be removed from the text, or some form of validation should be added (since pericentric siRNAs are abundant in WT cells, a northern blot for these should not be too challenging to perform).

Points 5-7 satisfactorily addressed.

(8) The decrease in incorporation of new H3 seen in *leo1Δ* seems relatively small, and at all but one of the loci tested could be explained as a consequence, rather than a cause, of increased heterochromatin (this is true for *Tf2*, *dhl*, *matK*, *IRC1R*, and also *rad50*, which lies just outside the right side of centromere one and therefore could be affected by heterochromatin spreading in *leo1Δ*). The authors should test some more sites that are not associated with increased heterochromatin in *leo1Δ* in order to confirm that this effect is independent of heterochromatin. Moreover, on its own I feel this data provides insufficient evidence for the key conclusion that *Leo1* destabilises heterochromatin by promoting histone turnover - this model should be tested by an alternative approach, for example testing whether other mutants affecting histone turnover (e.g. mutants of the HIR complex) also affect the stability of chromatin states, as would be predicted.

This point has been heavily discussed. As strongly suggested by the reviewer, we have now incorporated data on the interaction with two histone acetyltransferases that affect histone turnover Hat1 (in *S. cerevisiae*) and Mst2. Mst2 that has been shown to directly affect histone turnover in *S. pombe* (Wang et al, 2015, Elife). When deleting *mst2*, we find a similar stabilization of heterochromatin as with defect *leo1* or *paf1* (Fig 7D). The *hat1Δ* strain show no effect (Fig EV8). We discuss these differences in detail in the Discussion section (fourth paragraph, p9). We have also performed the RITE turnover measurements at two more euchromatic loci, (*pyk1* and *scm3*). We find the same effect that the histone turnover is reduced in *leo1Δ* cells (Fig 7C).

The effect of heterochromatin formation in HIRA mutants was previously determined in the Whitehall and Grewal labs (Anderson et al, 2010; Blackwell et al 2004; Yamane et al, 2011). Contrarily to the suggestion by the reviewer, HIRA proteins actually maintain the heterochromatin and mutations create a less condensed heterochromatin. We have now included a reflection on this in the Discussion section (p9: "During replication, Swi6 associates with a Clr6-HDAC complex and HIRA, which incorporate new histones that readily acquire the post-translational modifications of silent chromatin (Anderson et al, 2010; Blackwell et al, 2004; Yamane et al, 2011)"). For stringency, we no longer write "heterochromatin spreading" but heterochromatin stability" as this better reflects the data. We have added more references and more clearly discuss this in the current manuscript.

With these data we have now made a stronger case and we hope the reviewer agrees that it is the histone turnover as such that determines the observed heterochromatin stabilization.

The addition of RITE assay measurements for two more euchromatic loci has confirmed that loss of *Leo1* has a general effect on histone turnover at RNAPII-transcribed loci, independently of heterochromatin.

The conclusions that can be drawn from the new data on heterochromatin stabilisation in *mst2Δ* and *hat1Δ* cells are less clear cut. Although in the text the authors state that "inactivation of *Epe1*, *Leo1*, *Paf1* and *Mst2* have a similar effect on propagation of heterochromatin across IR-L", in fact Fig.EV8 shows that although an effect of *mst2Δ* on heterochromatin spreading at this locus can be seen, it is much weaker than the effect of *paf1Δ*. That said, I accept the point made by the authors in the discussion that different factors affecting histone turnover do so by different mechanisms, and these may have different outcomes in terms of heterochromatin stabilisation. Therefore, although I think that some of the statements on the *mst2Δ* phenotypes should be toned down, overall I am satisfied that the data now presented support the conclusion that heterochromatin stabilisation in cells lacking *Leo1* is likely due to reduced histone turnover.

Additional comments:

- (1) Despite improvements to some sections there are still issues with the clarity of the manuscript; in particular, there are several places where the logical flow, and conclusions drawn from, a set of experiments are still not clearly explained. For example, in the section on TRAMP/exosome, I still find it difficult to follow what point(s) the authors wish to make.
- (2) In Figure 4A and B, the quantitation of peaks that has been added is helpful, but it should be explained how these values were calculated e.g. do they represent maximum height of the peak, total area under the peak, or something else?

1st Revision - authors' response

27 September 2015

We would like to thank the reviewer for the suggestions and comments, which again have improved the study. Our current responses are in blue below.

Referee #1:

In the revised manuscript the authors have removed some of the original data that was difficult to interpret and contributed little to the overall story - I think this was a good idea as the focus and readability have improved as a result. The manuscript has also benefited from the addition of some

new data, in particular: (1) evidence showing that the heterochromatin stabilisation in *leo1Δ* cells reported here is distinct from the enhanced RNAi-mediated heterochromatin formation recently reported in *leo1Δ* cells by Kowalik et al; and (2) further data that somewhat strengthens the key conclusion of a causal link between histone turnover and heterochromatin spreading.

However, I have still have a number of concerns, including several points raised in my original review that have not been addressed and in my view are important to support the conclusions drawn. Comments on original major points:

(1) In relation to Fig.1, it is stated in the text that a *Leo1*1-169 allele was created to rule out secondary effects of transposon integration and that this mutant, like the *hermes* insertion, has a more severe phenotype than the *leo1Δ* null. However, no data on silencing/H3K9me2 in this mutant is presented - this data should be added.

We have now confirmed the unique integration by Southern blot (Figure EV1B). Due to the editor's concern for timeliness, we have not included ChIP-qPCR data for the *leo1*1-169 strain.

Without either a silencing assay or ChIP-qPCR data, the statement that this mutant has a more severe phenotype remains unsupported - the data on which this statement is based must be shown.

We have now included ChIP-qPCR data for the *leo1*1-169 strain (Fig 1C). This assay is largely qualitative (especially for H3K9me2) and we see the same levels of H3K9me2 in the mutant strains. We have changed the text to better reflect this. By “phenotype” we referred to the growth defect, which is now clarified (“the disrupted *leo1+* alleles (*leo1::Hermes* and *leo1*1-169::*HA*) exhibited a severe growth defect not present in the full gene deletion (*leo1Δ*)”) and shown in Fig 1E”).

(2) On p.5 it is stated that deletion of *cdc73* or *trp1* did not rescue the *ddb1Δ* phenotype, however, the data presented in Fig. EV2B are insufficient to support this conclusion as WT and *ddb1Δ* single mutant strains are not shown on the plate for comparison.

The growth of WT and *ddb1Δ* single mutant is shown in Fig 3A.

This is not sufficient as it is a different figure - to be able to properly assess whether *cdc3Δ* or *trp1Δ* rescue the *ddb1Δ* phenotype, the double mutants and the *ddb1Δ* single mutant must be compared side-by-side in the same assay.

We now provide the growth of the single and double mutants on the same plate (Fig EV2B).

(3) From the images shown in Fig. 3D I am not convinced that localisation of Epe1 is unaffected by deletion of *leo1*. This should be confirmed by chromatin-IP.

This is a good suggestion. However, due to the editor's concern regarding timeliness we have not performed this experiment, although it is technically feasible.

I think this experiment is important; based only on the images in Fig.3D I would not agree with the conclusion that deletion of Epe1 has no effect on Epe1 localisation.

As requested, we performed the ChIP of Epe1-GFP in WT, *paf1Δ* and *swi6Δ* strains. The results were not convincing, and would require more extensive experimentation to be conclusive on the Epe1 localization. Therefore we decided to omit this short section here. The conclusion related to the localization of Epe1, and by extension the disruption Epe1-Swi6, is not essential to the study. We are grateful to the reviewer for identifying this problem and helping us to strengthen the manuscript.

(4) Fig.5 - from the small RNA sequencing data it is inferred that siRNA levels are reduced in *leo1Δ* cells. However, since it is difficult to tell whether the apparent reduction reflects a genuine decrease in siRNA levels, or simply an increase in the proportion of non-siRNA RNAs sequenced, this finding should be confirmed by northern blot.

This is true. We don't know if the siRNAs are reduced, and they may not be. However, this is not essential for our main conclusion that heterochromatin is destabilized by Pac1C-promoted histone turnover. What is important is that the siRNAs are not increased as would have been expected from an RNAi-dependent increase in heterochromatin. We have rewritten the text.

Although this would be a nice experiment to do, in our experience Northern blots of siRNA are very cumbersome. These experiments would delay our paper significantly. We now put considerably less focus on the siRNA in the manuscript, as these were largely negative data.

Although the authors acknowledge that they cannot say whether siRNAs are reduced or not, the text still refers to siRNAs being reduced; for example: "...low levels of siRNAs were sufficient for heterochromatin nucleation or that an alternative heterochromatin nucleation mechanism compensated for the RNAi machinery". Either all suggestion that siRNAs are reduced should be removed from the text, or some form of validation should be added (since pericentric siRNAs are abundant in WT cells, a northern blot for these should not be too challenging to perform).

All suggestions of reduced siRNAs are removed. We have revised the above sentence to "...indicated that heterochromatin formation is intact, whether by RNAi or an alternative heterochromatin nucleation mechanism." There are no other references to levels of siRNAs from our data.

Points 5-7 satisfactorily addressed.

(8) The decrease in incorporation of new H3 seen in *leo1Δ* seems relatively small, and at all but one of the loci tested could be explained as a consequence, rather than a cause, of increased heterochromatin (this is true for *Tf2*, *dhI*, *matK*, *IRC1R*, and also *rad50*, which lies just outside the right side of centromere one and therefore could be affected by heterochromatin spreading in *leo1Δ*). The authors should test some more sites that are not associated with increased heterochromatin in *leo1Δ* in order to confirm that this effect is independent of heterochromatin. Moreover, on its own I feel this data provides insufficient evidence for the key conclusion that *Leo1* destabilises heterochromatin by promoting histone turnover - this model should be tested by an alternative approach, for example testing whether other mutants affecting histone turnover (e.g. mutants of the HIR complex) also affect the stability of chromatin states, as would be predicted.

This point has been heavily discussed. As strongly suggested by the reviewer, we have now incorporated data on the interaction with two histone acetyltransferases that affect histone turnover *Hat1* (in *S. cerevisiae*) and *Mst2*. *Mst2* that has been shown to directly affect histone turnover in *S. pombe* (Wang et al, 2015, *Elife*). When deleting *mst2*, we find a similar stabilization of heterochromatin as with defect *leo1* or *paf1* (Fig 7D). The *hat1Δ* strain show no effect (Fig EV8). We discuss these differences in detail in the Discussion section (fourth paragraph, p9).

We have also performed the RITE turnover measurements at two more euchromatic loci, (*pyk1* and *scm3*). We find the same effect that the histone turnover is reduced in *leo1Δ* cells (Fig 7C). The effect of heterochromatin formation in HIRA mutants was previously determined in the Whitehall and Grewal labs (Anderson et al, 2010; Blackwell et al 2004; Yamane et al, 2011). Contrarily to the suggestion by the reviewer, HIRA proteins actually maintain the heterochromatin and mutations create a less condensed heterochromatin. We have now included a reflection on this in the Discussion section (p9: "During replication, *Swi6* associates with a *Clr6*-HDAC complex and HIRA, which incorporate new histones that readily acquire the post-translational modifications of silent chromatin (Anderson et al, 2010; Blackwell et al, 2004; Yamane et al, 2011)").

For stringency, we no longer write "heterochromatin spreading" but heterochromatin stability" as this better reflects the data. We have added more references and more clearly discuss this in the current manuscript.

With these data we have now made a stronger case and we hope the reviewer agrees that it is the histone turnover as such that determines the observed heterochromatin stabilization.

The addition of RITE assay measurements for two more euchromatic loci has confirmed that loss of *Leo1* has a general effect on histone turnover at RNAPII-transcribed loci, independently of heterochromatin.

The conclusions that can be drawn from the new data on heterochromatin stabilisation in *mst2Δ* and *hat1Δ* cells are less clear cut. Although in the text the authors state that "inactivation of Epe1, Leo1, Paf1 and Mst2 have a similar effect on propagation of heterochromatin across IRL", in fact Fig.EV8 shows that although an effect of *mst2Δ* on heterochromatin spreading at this locus can be seen, it is much weaker than the effect of *paf1Δ*. That said, I accept the point made by the authors in the discussion that different factors affecting histone turnover do so by different mechanisms, and these may have different outcomes in terms of heterochromatin stabilisation. Therefore, although I think that some of the statements on the *mst2Δ* phenotypes should be toned down, overall I am satisfied that the data now presented support the conclusion that heterochromatin stabilisation in cells lacking Leo1 is likely due to reduced histone turnover.

As suggested, we have toned down the conclusions for the *mst2Δ* strain. In the Discussion section we now write "inactivation of Epe1, Leo1, Paf1 and, to a lesser extent, Mst2 have an effect on propagation of heterochromatin"

Additional comments:

(1) Despite improvements to some sections there are still issues with the clarity of the manuscript; in particular, there are several places where the logical flow, and conclusions drawn from, a set of experiments are still not clearly explained. For example, in the section on TRAMP/exosome, I still find it difficult to follow what point(s) the authors wish to make.

We have rewritten this section for clarity. The purpose of the TRAMP/exosome section is to compare how the *leo1/paf1* effects relate to previous studies of heterochromatin stability, especially mediated by siRNA-independent pathways. We have emphasized this point in the current manuscript.

(2) In Figure 4A and B, the quantitation of peaks that has been added is helpful, but it should be explained how these values were calculated e.g. do they represent maximum height of the peak, total area under the peak, or something else?

The values represent the RPKM (reads per kilobase per million mapped reads) or ratio thereof, for the regions in question. This has now been clarified in the legend for Figure 4: "The ratios of RPKM (reads per kilobase per million mapped reads) over the indicated region is shown.[...] The RPKM values are shown for each gene and strain."

2nd Editorial Decision

30 September 2015

Thank you for the submission of your newly revised manuscript. The referee is now happy with the study, and we can therefore in principle accept it.

I have seen that the manuscript has 9 expanded view figures now. Unfortunately, at the moment our publisher can only process 5 EV figures, and remaining EV figures need to be included in the Appendix file (please see our instructions to authors on our website for more information). Can you therefore please either combine EV figures in order to reduce their number to 5, or move the less relevant ones to an Appendix file, together with their legends. The EV tables can all remain. Please upload the 5 EV figures as individual files.

Please also upload together with the final manuscript the completed author checklist that I mentioned in my last letter, and that can be found at <http://embor.embopress.org/authorguide#revision>. Please insert page numbers in the checklist to indicate where in the manuscript the requested information can be found.

Regarding statistics, the figure legends do not mention the number "n" of how many independent experiments were performed and they do not yet define the bars and error bars, e.g. SEM or SD. Please add this information as it must be included in the figure legends, including the expanded view figure legends.

I would like to suggest a few changes to the abstract, which needs to be written in present tense:

The maintenance of open and repressed chromatin states is crucial for the regulation of gene

expression. To study the genes involved in maintaining chromatin states, we generated a random mutant library in *Schizosaccharomyces pombe* and monitored the silencing of reporter genes inserted into the euchromatic region adjacent to the heterochromatic mating type locus. We show that Leo1-Paf1 [a subcomplex of the RNA polymerase II associated factor 1 complex (Paf1C)] is required to prevent the spreading of heterochromatin into euchromatin by mapping the heterochromatin mark H3K9me2 using high-resolution genome-wide ChIP (ChIP-exo). Loss of Leo1-Paf1 increases heterochromatin stability at several facultative heterochromatin loci in an RNAi-independent manner. Instead, deletion of Leo1 decreases nucleosome turnover, leading to heterochromatin stabilization. Our data reveal that Leo1-Paf1 promotes chromatin state fluctuations by enhancing histone turnover.

Please let me know whether you agree with these changes.

EMBO reports papers are accompanied online by A) a short (1-2 sentences) summary of the findings and their significance, B) 2-3 bullet points highlighting key results and C) a synopsis image that is either exactly 211x157 pixels, or 550x200-400 pixels large. For the larger image the height is variable. You can either show a model or key data in the synopsis image. Please note that the smaller size is rather small and that text needs to be readable at the final size. Please send us this information along with the revised manuscript.

2nd Revision - authors' response

04 October 2015

Thank you for your positive response to our manuscript 'The Paf1 complex factors Leo1 and Paf1 promote local histone turnover to modulate chromatin states in fission yeast' (tracking number EMBOR-2015-41214V2). We have now revised the text to adhere to the guidelines of EMBO Reports.

3rd Editorial Decision

05 October 2015

I am very pleased to accept your manuscript for publication in the next available issue of EMBO reports. Thank you for your contribution to our journal.